# Alcohol Misuse: Integrating Personality Traits and Decision-Making Styles for Profiling

**DOI:** 10.3390/bs15050622

**Published:** 2025-05-02

**Authors:** Luis F. García, Lara Cuevas, Oscar García, Ferran Balada, Anton Aluja

**Affiliations:** 1Department Psicología Biológica y de la Salud, Facultad de Psicología, Universidad Autónoma de Madrid, 28049 Madrid, Spain; 2Lleida Institute for Biomedical Research, Dr. Pifarré Foundation, 25198 Lleida, Spain; oscar.garcia@universidadeuropea.es (O.G.); ferran.balada@uab.cat (F.B.); anton.aluja@udl.cat (A.A.); 3Cardenal Cisneros University College, 28006 Madrid, Spain; lacuevas@ucm.es; 4Department Psychobiology and Methodology in Behavioural Sciences, Universidad Complutense de Madrid, 28223 Madrid, Spain; 5Department Psychology, European University of Madrid, 28670 Villavicosa de Odon, Spain; 6Department Psicobiologia i Metodología CCSS, Facultad de Psicologia, Autonomous University of Barcelona, 08193 Barcelona, Spain; 7Department Psicología, Facultat de Psicología, University of Lleida, 25001 Lleida, Spain

**Keywords:** alcohol misuse, personality profiles, disinhibited personality, decision-making styles, sensation seeking, impulsivity, negative affect

## Abstract

The literature has described how different and independent personality profiles (pathways or motives) lead to the same outcome: alcohol misuse. In addition, decision-making styles could also play a role in understanding alcohol misuse better, although the evidence is much more scarce compared to personality traits. The present paper aims to test how personality traits and decision-making styles could be integrated to better understand different pathways/profiles of alcohol misuse. Measures of alcohol misuse (AUDIT and RAPI), structural personality models (ZKA-PQ/SF), impulsivity (BIS-11 and UPPS-P), and decision-making styles (GDMS) were applied to a sample of 988 individuals from the Spanish general population (446 of them also completed the NEO-PI-R). Exploratory factor analyses support the identification of different pathways to alcohol misuse, and regression analyses suggest that decision-making styles add little variance to personality traits to account for differences in alcohol misuse, although the spontaneous style is consistently associated with alcohol misuse. The conclusions highlight the need to consider different aetiologies of alcohol misuse, especially an antisocial/disinhibited profile, and claim for the assessment of decision-making styles and, especially, personality traits to facilitate more successful treatment and prevention programs for alcohol misuse.

## 1. Introduction

Alcohol misuse involves a progressive increase in time and amount of ethanol consumed, often together with a decline in non-alcohol-related activities. It results in mental and behavioural disorders and the development of cognitive deficits ([15]; [60]; [63]). Long-term abuse of alcohol can lead to neurological dysfunction that, in turn, has serious real-life consequences for individuals and societies ([29]). In fact, alcoholism is one of the major health problems in the world ([69]).

Considering the wide-ranging negative consequences of alcohol misuse within every human society and throughout the world, many research efforts have been devoted to predicting and deactivating it. Now, there is no doubt that alcohol misuse is associated with multiple biological, psychological, and social variables ([24]; [32]; [37]; [57]). From this standpoint, it presents a complex etiology that cannot be explained by one factor only.

### 1.1. Alcohol, Personality Traits, and Decision-Making Styles

One of the factors most associated with alcohol misuse has been personality traits (i.e., [6]; [46]; [44]; [61]). In a meta-analysis using the Five-Factor Model (FFM) as the personality framework, alcohol use disorder was associated with high neuroticism, low conscientiousness and low disinhibition ([38]). Later, and using an impressively large sample (72,949 adults), [31] ([31]) reported that high alcohol consumption was associated with high extraversion and low conscientiousness, whereas abstinence was associated with low extraversion, low openness and high agreeableness.

Considering other personality models, it is also usually observed that sensation seeking, disinhibition, impulsivity, and negative emotionality are strong predictors of alcohol consumption and problematic alcohol involvement ([6]; [30]; [52]). High levels of impulsivity and disinhibition would explain the capacity and willingness to adopt hasty, risky, and inappropriate behaviours or acts, such as alcohol misuse, which would result in unfavourable real-life consequences such as psychiatric disorders or economic deprivation ([35]; [39]).

Another relevant factor that has been researched is decision-making styles. Deficits in decision-making could affect how a person plans actions or impact self-control in the presence of dangerous but attractive stimuli ([22]), as is the case with alcoholic beverages. In this way, previous studies had been conducted to investigate the association between decision-making styles and alcohol use. Using the Melbourne Decision Making Questionnaire (MDMQ, [45]), [51] ([51]) reported that lower vigilance and high procrastination scores were associated with a slightly greater risk of alcohol misuse. Applying the General Decision-Making Style (GDMS; [59]), [12] ([12]) reported that avoidant (β = 0.23) and spontaneous (β = 0.27) decision-making styles were related to a higher level of problems related to alcohol use. Later, [10] ([10]) explored the combined effect of personality traits and decision-making styles to understand several health problems in a sample of Slovakian university students. They reported that decision-making styles marginally improved the prediction of alcohol problems reached by personality traits, but they also suggested that dependent and avoidant decision-making styles could play some role.

Convergent validity and theoretical studies show that procastination ([33]; [64]) and dependent and avoidant ([59]; [65]) decision-making styles are mainly associated with the neuroticism personality trait. Therefore, deficits in control of negative emotions ([43]) seem to play the main role in the association between decision-making styles and alcohol misuse. As a final remark, considering that the vigilance style (MDMQ) presents the highest relationship with the rational style (GMDQ; [4]), it is surprising that the former was associated with alcohol misuse ([51]), whereas the latter was not ([10]).

### 1.2. Etiological Pathways for Alcohol Misuse

Several different profiles of alcohol use and misuse have been described (for a comprehensive review, see [46]). For instance, [70] ([70]) proposed a developmental model with four alcoholism subtypes: antisocial, developmentally limited, negative affect, and primary. Cloninger devised a well-known typology of alcoholism that has inspired much research on the topic ([17]). He suggested the presence of two types of alcoholics (I and II). Type I is characterized mainly by the motivation to reduce tension and anxiety, which results in a rapidly developed tolerance and psychological dependence. This type has been associated with the trait poles of high harm avoidance (i.e., high neuroticism) and low novelty seeking. On the other hand, Type II was associated with positive reinforcement for its euphoriant and stimulant effects because of their natural need of stimulation. This second type would present an antisocial and sensation-seeking profile. These and other typologies and classifications clearly suggest that people develop a substance disorder, addiction, or alcohol misuse due to different motivations and for different, mostly unrelated, etiologies ([21]).

In a review article, [66] ([66]) identified three pathways to substance abuse disorder and addictive behavior: (I) behavioural disinhibition; (II) stress reduction; and (III) reward sensitivity. Different traits underline these three pathways: antisocial, impulsive, disinhibited, and aggressive traits in the first case, neuroticism and anxiety in the second, and sensation seeking, reward dependence, and extroversion in the third. The impulsivity and sensation-seeking pathways, or the externalizing pathways, suggest that alcohol use forms part of a more general pattern of problematic or antisocial behavior, in some cases driven by fun-seeking. The second pathway, negative-affect regulation, also known as the self-medication or internalizing pathway, refers to drinking alcohol to decrease distress. It is important to highlight that these three pathways address personality traits and are expected to be independent, the presence of one of them being enough to develop alcohol misuse. Hence, considering all three together is essential to understand the specific motivation of people with an alcohol misuse condition.

Recently, [46] ([46]) proposed an alcohol developmental model. Interestingly, they explored the role of three broad personality domains (disinhibition, positive emotionality, and negative emotionality) in three different phases of alcohol intake (onset, use, misuse). Disinhibition would be largely involved in the three phases and plays a prominent role to account for use and misuse of alcohol. Positive emotionality would be associated with the beginning of alcohol consumption and specific patterns of heavy consumption, but its impact would be less influential in the development of long-term alcohol misuse, and, finally, negative emotionality would be mainly involved in alcohol misuse. This model is supported by some results that suggest that excessive alcohol consumption during adolescence may be partly driven by excitement seeking, but problematic use may be a consequence of disinhibition or an attempt to reduce negative mood ([62]).

### 1.3. Aims

The literature largely supports that personality traits and decision-making styles are associated with alcohol misuse, the former being one of the most important factors to understand the predisposition, continuous use, and serious negative real-life consequences of alcohol misuse. So, the first aim is to replicate the reported relationships between alcohol misuse and personality traits and decision-making styles. It is expected that the aggressiveness, impulsivity, sensation-seeking, and neuroticism traits will be associated with alcohol misuse. What is difficult to predict is the specific decision-making styles that will play a relevant role given the inconsistencies in the previous literature, where different studies ([10]; [12]; [51]) have emphasized the role of different decision-making styles.

The second aim focuses on incremental validity. Previous studies have suggested that personality traits and decision-making styles are related. Some traits (such as neuroticism, conscientiousness, or sensation seeking) present correlations between 0.40 and 0.60 with decision-making styles (i.e., [33]; [53]; [65]). Hence, given the high overlapping between the two constructs, we will test how much incremental variance is added to the other kind of construct and also combine the two to try to predict alcohol misuse as accurately as possible. It is hypothesized that decision-making styles will add some incremental validity beyond personality traits, but not much ([10]).

Furthermore, it is compulsory to identify different pathways to alcohol misuse because not considering heterogeneity can severely bias its assessment and lead to ineffective treatment ([27]). Therefore, the third and main aim of the present paper is to describe personality profiles (i.e., motivations) of alcohol misuse according to the different pathways proposed by [66] ([66]) and [46] ([46]) and to test how much predictive power decision-making styles add for every profile. In order to describe different pathways as broadly and accurately as possible, several personality measures (Zuckerman’s personality model, impulsivity and FFM) were applied. Considering [46]’s ([46]) model, it is expected that the disinhibition pathway will present the largest association with alcohol misuse, although reward sensitivity (i.e., sensation seeking) and negative emotionality (i.e., neuroticism) will be associated with alcohol misuse as well.

## 2. Materials and Methods

### 2.1. Participants

A total of 988 individuals (46.7% males and 53.3% females) from the Spanish general population participated in the present study. The mean age was 44.52 years (s.d. = 19.87) with a minimum of 18 and a maximum of 91. Briefly, 276 (27.9%) were university students, 359 (50.4%) worked for a large or a medium-sized company, and the rest were specialized, semi-specialized, or unskilled workers. All of them completed a protocol that included measures of alcohol misuse, Zuckerman’s personality and impulsivity models, and decision-making styles. In addition, 446 of them also completed the NEO-PI-R.

### 2.2. Instruments

#### Alcohol Misuse

Two measures of alcohol misuse were collected: The Alcohol Use Disorders Identification Test (AUDIT) and the Rutgers Alcohol Problems Index (RAPI). The AUDIT is a 10-item screening tool developed by the World Health Organization (WHO) to assess alcohol consumption, drinking behaviours, and alcohol-related problems ([58]). Responses to each question are scored from 0 to 4, giving a maximum possible score of 40. The value of 0 indicates an abstainer who has never had any problems with alcohol. A score of 1 to 7 suggests low-risk consumption according to World Health Organization (WHO) guidelines. Scores from 8 to 15 suggest hazardous or harmful alcohol consumption and a score of 16 or more indicates the likelihood of alcohol dependence (moderate/severe alcohol use disorder). In the present sample, two groups were formed: low risk (891 in the total sample and 406 in the subsample completing the NEO-PI-R) and high risk (hazardous and alcohol dependence groups; 97 in the total sample and 40 in the subsample completing the NEO-PI-R). The Spanish adaptation of AUDIT was developed by [56] ([56]). The RAPI is a reliable and valid instrument for detecting alcohol-related problems. It was initially developed by [67] ([67]) and consists of 23 items with a Likert-type response format of 0 to 3. Participants stated how many times a certain alcohol-related event had occurred in their life over the past year, marking 0 if it had never happened, 1 if it had happened once or twice, 2 if it had occurred 3–5 times, and 3 if it had occurred more than five times. It was adapted to Spanish by [42] ([42]), reporting excellent psychometric properties. It should be highlighted that this instrument was initially developed for young populations, but we also applied it to adults in the present study.

### 2.3. Personality Measures

*Zuckerman–Kuhlman-Aluja Personality Questionnaire shortened form* (*ZKA-PQ/SF*; [7]). The ZKA-PQ/SF is a short version of the ZKA-PQ that measures five personality domains: aggressiveness (AG), activity (AC), extraversion (EX), neuroticism (NE), and sensation seeking (SS). This is an abbreviated version (80 items) of the longer original ZKA-PQ ([5]). The response format is a 4-point Likert-type scale ranging from 1 (strongly disagree) to 4 (strongly agree). Validity and reliability evidence of the ZKA-PQ/SF are appropriate according to the cross-cultural data presented in various African, American, Asian, and European cultures and languages ([7], [8]; [55]).

*Impulsivity measures.* In this study, two impulsivity questionnaires were used: The BIS-11 ([50]) and UPPS-P ([68]). The Barratt Impulsiveness Scale (BIS-11) is a 30-item questionnaire that taps three scales: attention (AI), motor (MI), and non-planning (NPI) impulsiveness. It was adapted to the Spanish cultural context by [49] ([49]). The answer format has a 4-point scale ranging from 1 to 4 (rarely/never, occasionally, often, almost always). A total score can be computed as well, and it was the only score used in the present study. The Impulsive Behavior Scale (UPPS-P) shortened version contains 20 items and five scales: Negative Urgency, Lack of Premeditation, Lack of Perseverance, Sensation Seeking and Positive Urgency. The items are scored on a four-point Likert scale, ranging from 1 (strongly agree) to 4 (strongly disagree). The Spanish version was validated by [16] ([16]).

*The Revised NEO Personality Inventory (NEO-PI-R).* It is a well-known measure of the FFM: neuroticism (N), extraversion (E), openness to experience (O), agreeableness (A), and conscientiousness (C). The 240 items of the questionnaire are answered on a 5-point Likert-type scale (0–4), ranging from “strongly disagree” to “strongly agree”. The Spanish version of the NEO-PI-R ([19]) has good psychometric properties, similar to those of the original American version.

### 2.4. Decision-Making Styles

*General Decision-Making Scale* (*GDMS*; [59]). The GDMS is a self-administered instrument originally designed by [59] ([59]) with 25 items. [2] ([2]) adapted it to Spanish, removing three items (resulting in a total of 22) based on previous studies (e.g., [12]) and their own analysis. It was structured by five different domains, each representing a decision-making style (number of items between brackets): rational (5), intuitive (3), dependent (5), avoidant (5), and spontaneous (4). The rational decision-making style involves the use of reasoning, logical, and structured approaches to decision-making. The intuitive decision-making style is defined by reliance upon hunches, feelings, impressions, instinct, and good feelings. The dependent style is defined by a search for advice and guidance from others before making important decisions. The avoidant decision-making style is defined by withdrawing, postponing, and moving back and negating the decision scenarios. A spontaneous style is characterized by a feeling of immediacy and a desire to get through the decision-making process as quickly as possible. The response format consists of a 5-point Likert scale ranging from 1 (strongly disagree) to 5 (strongly agree).

### 2.5. Procedure

Undergraduate psychology students were trained in the application of psychological instruments. As a regular exercise, they had to administer a protocol containing the instruments analysed in the present study, as well as other psychological instruments to seven people with the following characteristics: the student her/himself, one male and female aged between 18 and 30 years, one male and female aged between 31 and 50 years, and one male and female older than 51 years. It should be remarked that neither the name nor a personal identification number (such as identity card) nor other personal information was recorded. The research was part of a project authorized by the university of Lleida ethics committee (Code: CEIC 2160). All participants were informed that anonymous data could be used for research purposes. No reward was given for completing the protocol, but, to increase motivation, scores on some personality traits were returned to all participants. The database and all information about this study was saved in a computer protected by a personal password. The handling of the information has been carried out in accordance with the confidentiality rules set out in the Spanish Organic Law 3/2018 on Data Protection and Guarantee of Digital Rights, the Helsinki Declaration, and the Council of Europe Convention on Human Rights.

### 2.6. Data Analysis

Firstly, correlations of personality scales and decision-making styles with both alcohol misuse instruments were computed. A complementary non-linear analysis was also conducted to explore the impact of every personality and decision-making style on alcohol misuse. In this way, Cohen’s d of the differences between the low- and high-risk groups (AUDIT) was also computed. Secondly, in order to test the incremental validity of both personality traits and decision-making styles over the other kind of variable, two hierarchical regression analyses were performed, introducing the scales of one kind of variables in the first block and the other in the second. The enter method was used. This analysis was conducted for both dependent variables (AUDIT and RAPI).

To describe the personality profiles (pathways) of alcohol misuse, a factor analysis was conducted introducing the ZKA-PQ/SF, BIS-11, and UPPS-P scales. Considering the proposals by [66] ([66]) and [46] ([46]), three factors were requested. It is expected to identify antisocial/aggressiveness, impulsivity, and sensation-seeking factors in the total sample. Factors were extracted using principal components and rotated with the Oblimin procedure. It is necessary to use an oblique rotation method to test if the three profiles are independent or present a high overlapping. Negligible correlations are expected, indicating different pathways. It is possible that the negative affectivity factor was not identified in the total sample because there is one scale associated with this trait only (Neuroticism scale of the ZKA-PQ/SF). Hence, the same analysis was conducted on the subsample including the NEO-PI-R to better identify the negative emotionality or stress reduction pathway. Factor scores for both factor analyses were computed using the regression method, and Cohen’s d of the differences among the low- and high-risk groups was also computed for these factor scores.

Finally, a hierarchical regression analysis was performed introducing factor scores (as a measure of pathways) as the initial predictive variable in a first block and the decision-making styles in the second block (enter method). Thus, we tested the incremental validity of the decision-making styles for every pathway in regard to alcohol misuse. This analysis was also conducted for both AUDIT and RAPI.

## 3. Results

### 3.1. Association of Alcohol Misuse with Personality Traits and Decision-Making Styles

Table 1 shows the mean, standard deviations, skewness, and kurtosis of all scales in the present sample. As expected, all scales show normal distribution but both scales of alcohol misuse present, as usual, a positive deviation from normality. Hence, most of the people do not present serious alcohol use problems. Reliabilities were also adequate. Correlations and effect sizes indicate that impulsivity (also conscientiousness), sensation-seeking, and antisocial traits (aggressiveness, agreeableness, and the previously mentioned conscientiousness) present relevant associations with alcohol misuse. Regarding decision-making styles, avoidant and spontaneous present the largest associations (Table 2). Correlations among all scales analysed and with age and sex are reported in Appendix A.

### 3.2. Incremental Validity of Personality Traits and Decision-Making Styles to Account for Alcohol Misuse

When the two kinds of variables are combined to predict alcohol misuse, they account for about 20% of the variance (Table 3). Results also show that personality traits are a good deal more relevant than decision-making styles, since the latter do not add predictive variance and also account for less than 8% when they are entered in the first block. Note that results were almost equivalent across both measures of alcohol misuse. A further analysis was conducted introducing age and sex in the first block, personality scales in the second, and decision-making styles in the third. Age accounted for about 10% of the variance of alcohol misuse, personality traits added about another 10%, and no decision-making style was associated with alcohol misuse. As the same personality scales were entered in the final equation for both AUDIT and RAPI, this analysis replicates the pattern reported in Table 3.

### 3.3. Pathways to Alcohol Misuse: Profiling, Prediction, and Incremental Validity of Decision-Making Styles

For profiling the personality pathways, a factor analysis in the total sample was computed. For theoretical reasons ([46]; [66]), three factors were extracted (Table 4), but it should be noted that the Kaiser rule and Scree test suggest three factors as well. The three factors resemble the pathways described in the literature ([46]; [66]): one factor is close to antisocial tendencies (Aggressiveness), the second one includes the sensation-seeking scales, and the third one is clearly defined by the impulsivity scales (three subscales of the UPPS-P, BIS-11, and activity in negative). It must be highlighted that the three factors were mostly uncorrelated amongst each other. Cohen’s d shows medium effect sizes but somewhat larger than those reported for the scales individually (Table 2). It is also essential to remark that the inclusion of one neuroticism scale only precludes the identification of stress reduction ([66]) or negative emotionality ([46]) pathways. In fact, in this solution the neuroticism scale of the ZKA-PQ/SF loaded on the first factor (antisocial/aggressiveness).

When the regression analyses including the factor scores as independent variables are conducted (Table 5), the percentage of variance is about 10% for aggressiveness/neuroticism, 7% for sensation seeking, and 3% for impulsivity. It seems that the antisocial pathway presents the largest correlation with alcohol misuse. It should be remarked that decision-making styles increment the predictive validity by about 5%, the most important styles being once again avoidant and spontaneous. Like the previous analysis, no sharp differences are reported between results for AUDIT or RAPI.

Considering the potential role of neuroticism or negative emotions in understanding alcohol misuse, a new factor analysis introducing the NEO-PI-R domains was also computed to force the extraction of a neuroticism factor. Table 6 shows the loadings on the four-factor solution. A fourth factor was extracted to add the neuroticism factor. As expected, the solution reproduces the three factors described in Table 4 but the aggressiveness/neuroticism one is split in two factors: aggressiveness and neuroticism. Also, as expected, the four factors were independent, and the effect sizes were also medium, except for neuroticism, which was small. Table 7 shows a hierarchical regression analysis introducing successively the factor scores of the four pathways as independent variables in the first block and decision-making styles in the second. Results show low percentages of variance, the highest being for the aggressiveness pathway once again. Once again, the accounted variance for the decision-making styles was low, but higher for the neuroticism factor (8%), with the spontaneous styles playing the main role.

## 4. Discussion

Personality traits and decision-making styles are associated with alcohol misuse, and, as expected, the former accounted for more variance in alcohol misuse (with percentages between about 10 and 20%; [6]) than the latter. In detail, the results are in strong agreement with the literature since disinhibited/antisocial, impulsivity, sensation-seeking, and negative emotional traits are associated with alcohol misuse ([6]; [30]; [41]). Other specific traits such as conscientiousness or agreeableness also showed an association with alcohol misuse.

The present study largely supports the definition of different profiles/pathways (i.e., motivations) to alcohol misuse. It should be highlighted that all pathways were related to alcohol misuse, although they were uncorrelated. It therefore largely reinforces the need to explore different pathways to alcohol misuse. As we also expected, the antisocial/disinhibition pathway is the most important profile to understand misuse, which clearly supports the developmental model put forward by [46] ([46]). People that tend to be aggressive and disrespectful of social norms would have the highest risk of developing alcohol misuse and the corresponding negative outcomes in real life ([14]; [46]). Sensation seeking, meanwhile, would be the second most relevant pathway to alcohol misuse. In this case, motivation to consume could be attributable to appetitive motivation. This is also in agreement with the fact that enhancement drinking motives (i.e., “to get a high” or “because it’s fun”) have been associated with different alcohol-related outcomes, such as drinking quantity and frequency. The relevance of sensation seeking suggests that personality models other than the Big Five (which only considered sensation seeking as an extroversion facet in the gold standard of the FFM – NEO-PI-R; [19]) should be considered, in particular Zuckerman’s model, in which sensation seeking is a basic dimension of human personality ([71]). Note that this trait has been related to biases in decision-making ([54]), especially with unplanned and risky behaviours with negative consequences for the person and the group ([35]; [39]). Impulsivity, which is factorially and psychologically different from sensation seeking ([26]), also plays a role, albeit lesser than suggested in other studies ([1]).

Finally, a factor of neuroticism presents a lower predictive validity of alcohol misuse. This result was unexpected because negative emotionality or related traits have been strongly associated with the continuous pattern of alcohol use and misuse ([17]; [46]). Results of the present study could indicate that the association between negative emotionality and alcohol misuse might be due to antisocial aspects of neuroticism such as anger ([36]) more than anxiety itself. This argument would agree with [6]’s ([6]) results, where anxiety was not predictive of alcohol drinking, and with the fact that the association with alcohol misuse was higher for the neuroticism scales of the NEO-PI-R than for the corresponding scale of the ZKA-PQ/SF. Since the neuroticism domain of the NEO-PI-R includes facets of depression or anger ([19]), this pattern of results would indicate that alcohol misuse as a self-medication strategy would have other psychological motivations beyond stress reduction. It could be considered a strategy to reduce any negative emotion, such as sadness, anger, or undesired aggressivity ([25]).

The results support the hypothesis that, beyond personality traits, decision-making styles make a small contribution to understanding alcohol misuse (usually lower than 5%; [10]). In the present study, the spontaneous style plays the main role (as it does in Bavolar and Orosová, 2014). It should be highlighted that this decision-making style increases the predictive power irrespective of the profiling ([28]). Thus, the desire to resolve a situation quickly, though in an inappropriate or maladaptive way, puts a person at greater risk of developing alcohol misuse beyond the specific personality pathway. The main role of the spontaneous style is also in agreement with the fact that it is involved in other risky behaviours such as driving unsafely ([3]). In regard to the other decision-making styles, the avoidant style also seems to be relevant ([10]), but it is surprising that rational or intuitive, which have been associated with other negative mental health outcomes ([3]; [11]), played no role, although this pattern has already been replicated in previous studies ([10]; [12]).

Decision-making styles increased the predictive validity of all pathways, but especially so in the case of the neuroticism profile (8%). This would be in line with the need to present an anxious-impulsive tendency to understand the causes of the connection between negative emotionality and alcohol misuse. A fast and maladaptive decision-making style, possibly triggered by anxiety, would be necessary to explain why people high on neuroticism abuse alcohol ([9]). A tentative explanation of why decision-making styles increment the predictive power of all pathways could be the shared variance with other personality traits not considered in each profile ([13]; [64]). However, the fact that the spontaneous decision-making style was the best predictor irrespective of the profile, and that some decision-making styles related with relevant personality traits (i.e., with conscientiousness or neuroticism) do not play a role, goes against this argument and points to a genuine, albeit slight, association between decision-making styles and alcohol misuse.

According to the main aim of the present study, the idea of considering different pathways in the assessment and treatment of alcohol misuse is reinforced ([17]) or substance abuse disorder ([66]). Psychological interventions to reduce alcohol misuse ([48]) would benefit from considering personality traits as trigger factors of alcohol use and, especially, abuse. Different personalities shed light on understanding the specific motives of individuals for consuming alcohol and developing behavioural problems related to this use ([34]) and understanding better the decision-making dynamics that lead to negative clinical conditions such as alcohol misuse or other substance abuse disorders ([40]).

From a preventive standpoint, personality could make a big contribution to detecting high-risk individuals and designing selective and more effective programs for every individual to reduce and prevent negative alcohol-related outcomes ([23]). In this way, detecting disinhibited personality profiles would be one of the keys to detecting and preventing long-term negative consequences of alcohol misuse. For instance, identifying situational triggers of a disinhibited behavior or cognitive distortions associated with, for instance, the reinforcement of aggressive behavior would avoid serious alcohol misuse and, in turn, reduce the impact of alcohol on behavior ([18]; [47]). Focusing on these high-risk individuals would be a possibly more successful strategy to reduce the general alcohol use in a society and negative outcomes related to alcohol, which, in turn, would reduce costs for society and healthcare ([20]).

There are limitations of the present study which have implications for future studies. The cross-sectional study design is a limitation to disentangle the dynamic relationship between personality and decision-making styles and hence understand misuse. Longitudinal studies are needed to understand how a personality profile and decision-making styles combine over time to understand the beginning of alcohol use and how, for some individuals, this use becomes problematic. Moreover, since the present manuscript focuses on alcohol misuse measures, we can draw no conclusion about the beginning of alcohol use, and it has been argued that different personality factors affect different phases of alcohol consumption ([46]). Another limitation is the way to define the pathways. In the present study, only theoretical guided personality profiles have been defined. However, more accurate measures of the dynamics of alcohol use of individuals are necessary. For instance, a non-personality external assessment is required to demonstrate if heavy drink behaviour in one individual has the motivation of involvement in aggressive and disinhibited situations, stress reduction, or simply fun seeking ([66]). Finally, the present manuscript seeks to contribute to the design of successful interventions to reduce and prevent alcohol misuse. An essential line of research would be to test how psychological interventions customized according to personality pathways could improve the effectiveness of therapy focusing on reducing disinhibition or aggressive behaviours or negative mood states.

## 5. Conclusions

Alcohol misuse depends on multiple factors. In agreement with all the literature, the present study shows that one of the most important is personality traits, since this factor accounts for about 20% of the variance in alcohol misuse. It has also been reported that decision-making styles could be combined with personality traits to understand alcohol misuse, although their predictive power is lower. Finally, the main contribution of the present paper is to show that different personality profiles (pathways), although uncorrelated, are associated with alcohol misuse ([66]). In particular, the antisocial/aggressive/disinhibited profile presents the largest risk to present an alcohol misuse pattern, which supports the developmental model proposed by [46] ([46]). In summary, it is suggested that personality traits and decision-making styles, though mainly the former, should be routinely assessed to optimize and increase the efficacy of psychological interventions or public programs aimed at reducing alcohol use and misuse.

## Figures and Tables

**Table 1 behavsci-15-00622-t001:** Statistical descriptives and Cronbach’s alpha of alcohol misuse, ZKA-PQ/SF, BIS-11, UPPS-P, NEO-PI-R, and GDMS scales.

Scale	N	Mean	Standard Deviation	Skewness	Kurtosis	Alpha
AUDIT	988	3.39	3.54	2.08	5.62	0.76
RAPI	988	26.75	7.21	3.24	13.65	0.92
Aggressiveness	988	32.59	9.04	0.49	−0.13	0.89
Activity	988	41.18	7.60	0.22	−0.04	0.82
Extraversion	988	48.76	7.81	−0.47	0.00	0.85
Neuroticism	988	34.30	9.67	0.43	−0.29	0.90
Sensation Seeking	988	37.27	8.89	0.10	−0.59	0.84
Impulsivity (BIS-11)	988	62.21	9.33	0.36	0.12	0.78
Negative Urgency	988	8.51	2.73	0.16	−0.50	0.75
Lack of Premeditation	988	7.38	2.27	0.52	0.36	0.72
Lack of Perseverance	988	6.82	2.48	0.93	0.96	0.78
Sensation Seeking	988	8.49	3.11	0.33	−0.66	0.83
Positive Urgency	988	9.06	2.39	0.15	−0.28	0.60
Neuroticism	446	87.68	26.33	0.31	−0.11	0.91
Extroversion	446	105.70	22.17	−0.23	−0.21	0.86
Openness	446	109.02	22.72	−0.01	−0.11	0.87
Agreeableness	446	124.82	19.49	−0.29	−0.15	0.85
Conscientiousness	446	125.73	23.01	−0.30	−0.15	0.89
Rational	988	3.94	0.63	−0.68	0.76	0.82
Intuitive	988	3.67	0.80	−0.46	0.12	0.85
Dependent	988	3.38	0.83	−0.22	−0.02	0.83
Avoidant	988	2.39	0.97	0.51	−0.43	0.91
Spontaneous	988	2.26	0.87	0.50	−0.21	0.87

**Table 2 behavsci-15-00622-t002:** Correlations with both alcohol misuse scales and effect sizes (Cohen’s d) of the differences between low- and high-risk groups of alcohol misuse (AUDIT).

	Audit	Rapi	Cohen’s d
Aggressiveness	0.25	0.28	**0.69**
Activity	−0.09	−0.02	−0.14
Extraversion	0.01	−0.06	−0.01
Neuroticism	0.10	0.20	0.39
Sensation Seeking	**0.32**	**0.31**	**0.69**
Impulsivity (BIS-11)	**0.30**	**0.32**	**0.75**
Negative Urgency	0.16	0.24	**0.53**
Lack of Premeditation	0.18	0.19	0.49
Lack of Perseverance	0.13	0.19	0.44
Sensation Seeking	0.24	0.28	0.48
Positive Urgency	0.27	**0.30**	**0.81**
Neuroticism	0.16	0.24	**0.50**
Extroversion	0.15	0.13	0.45
Openness	0.14	0.14	0.45
Agreeableness	−0.21	−0.26	**0.62**
Conscientiousness	−0.29	−0.26	**0.67**
Rational	−0.12	−0.17	0.35
Intuitive	−0.03	−0.05	0.06
Dependent	−0.01	−0.01	0.05
Avoidant	0.14	0.20	**0.50**
Spontaneous	0.21	0.24	0.46

Correlations higher than ±0.30 and medium effect sizes (>0.50) are in boldface.

**Table 3 behavsci-15-00622-t003:** Hierarchical regression analysis for the total sample introducing personality scales (ZKA-PQ/SF, BAS-11, and UPPS-P) and decision-making styles for both alcohol misuse instruments.

	Block 1: Decision-Making Styles			Block 2: Personality				Block 1: Personality			Block 2: Decision-Making Styles			
Alcohol	R	R^2^_adj_	Scales	R	R^2^_adj_	Scales	ΔR^2^_adj_	R	R^2^_adj_	Scales	R	R^2^_adj_	Scales	ΔR^2^_adj_
AUDIT	0.23	0.05	SPO+	0.42	0.17	SS+, AC−, PU+, AG+	0.15	0.42	0.17	SS+, AC−, PU+, AG+	0.42	0.17		0.01
RAPI	0.29	0.08	AVO+, SPO+	0.44	0.18	SS+, PU+	0.11	0.43	0.18	SS+, PU+	0.44	0.18		0.01

SPO: Spontaneous; AVO: Avoidant; SS: Sensation Seeking, AC: Activity; PU: Positive Urgency; AG: Aggressiveness. The sign (+ or −) represents the direction (positive or negative, respectively) of the association.

**Table 4 behavsci-15-00622-t004:** Three-factor structure of the personality variables (profiling/pathways) in the total sample, correlations among factors, and Cohen’s d (last row) for the factor scores comparing low- and high-risk groups of alcohol misuse (AUDIT).

	Aggressiveness/Neuroticism	Sensation Seeking	Impulsivity
Aggressiveness	**0.776**	0.232	0.148
Activity	0.101	0.334	**−0.616**
Extraversion	−0.385	**0.513**	−0.144
Neuroticism	**0.723**	−0.157	0.172
Sensation Seeking	0.201	**0.839**	0.118
Impulsivity (BIS-11)	**0.523**	**0.551**	**0.523**
Negative Urgency	**0.773**	0.255	0.214
Lack of Premeditation	0.278	0.390	**0.747**
Lack of Perseverance	0.335	0.184	**0.817**
Sensation Seeking	0.231	**0.792**	0.141
Positive Urgency	**0.580**	**0.510**	0.241
	1.00		
	0.163	1.00	
	−0.238	−0.085	1.00
	**0.73**	**0.61**	0.48

Loadings higher than ±0.40 and medium effect sizes (>0.50) are in boldface.

**Table 5 behavsci-15-00622-t005:** Hierarchical regression analysis of the personality profiles and decision-making styles in the total sample.

	R	R^2^_adj_	Block 1: Profile	R	R^2^_adj_	Block 2: Styles	ΔR^2^_adj_
AUDIT	0.23	0.05	Aggressiveness/Neuroticism	0.28	0.07		0.02
RAPI	0.33	0.11	Aggressiveness/Neuroticism	0.37	0.13	SPO+	0.03
AUDIT	0.27	0.07	Sensation Seeking	0.31	0.09	AVO	0.03
RAPI	0.27	0.07	Sensation Seeking	0.35	0.12	AVO+, SPO+	0.05
AUDIT	0.18	0.03	Impulsivity	0.25	0.06	SPO+	0.03
RAPI	0.19	0.03	Impulsivity	0.30	0.08	AVO+, SPO+	0.05

SPO: Spontaneous; AVO: Avoidant. The signs (+ or −) represent the direction (positive or negative, respectively) of the association.

**Table 6 behavsci-15-00622-t006:** Four-factor structure of the personality variables (profiling/pathways) in the subsample including the NEO-PI-R, correlations among factors, and Cohen’s d in the last row for the factor scores comparing low- and high-risk groups of alcohol misuse (AUDIT).

	Aggressiveness	Sensation Seeking	Impulsivity	Neuroticism
Aggressiveness	**0.730**	−0.060	0.050	0.315
Activity	0.320	0.323	**0.594**	−0.020
Extraversion	−0.255	**0.698**	0.014	−0.283
Neuroticism	0.049	−0.062	0.004	**0.918**
Sensation Seeking	0.277	**0.710**	−0.125	−0.054
Impulsivity (BIS-11)	0.313	0.313	**−0.547**	0.061
Negative Urgency	**0.643**	−0.001	−0.064	0.231
Lack of Premeditation	0.142	0.158	**−0.777**	−0.169
Lack of Perseverance	−0.008	0.017	**−0.779**	0.074
Sensation Seeking	0.356	**0.583**	−0.089	−0.035
Positive Urgency	**0.496**	0.234	−0.190	0.114
Neuroticism	0.137	−0.066	−0.090	**0.869**
Extroversion	−0.033	**0.833**	0.134	−0.197
Openness	−0.289	**0.700**	−0.040	0.343
Agreeableness	**−0.761**	0.152	0.079	0.146
Conscientiousness	−0.133	0.012	**0.800**	−0.137
	1.00			
	0.129	1.00		
	−0.237	−0.059	1.00	
	0.193	−0.073	−0.216	1.00
	**0.70**	**0.62**	**0.50**	0.36

Loadings higher than ±0.40 and medium effect sizes (>0.50) are in boldface.

**Table 7 behavsci-15-00622-t007:** Hierarchical regression analysis of the personality profiles and decision-making styles in the subsample including the NEO-PI-R.

	R	R^2^_adj_	Block 1: Profile	R	R^2^_adj_	Block 2: Decision Making Styles	ΔR^2^_adj_
AUDIT	0.25	0.06	Aggressiveness	0.32	0.10	SPO+	0.04
RAPI	0.34	0.11	Aggressiveness	0.39	0.14	AVO+, SPO+	0.03
AUDIT	0.21	0.04	Sensation Seeking	0.33	0.10	SPO+	0.03
RAPI	0.23	0.05	Sensation Seeking	0.36	0.12	AVO+, SPO+	0.08
AUDIT	0.23	0.05	Impulsivity	0.31	0.10	SPO+	0.05
RAPI	0.21	0.04	Impulsivity	0.32	0.09	SPO+	0.06
AUDIT	0.11	0.01	Neuroticism	0.30	0.08	SPO+	0.08
RAPI	0.21	0.04	Neuroticism	0.35	0.11	SPO+, DEP+	0.08

SPO: Spontaneous; AVO: Avoidant; DEP: Dependent.

## Data Availability

Full data will not be available until the project is completed. However, partial data can be requested from the first and last authors.

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
