# Peer review of "Alcohol Misuse: Integrating Personality Traits and Decision-Making Styles for Profiling"

_behavsci, 2025, doi:10.3390/bs15050622_

Round 1
Reviewer 1 Report
Comments and Suggestions for Authors
This article aims to explore how personality traits and decision-making styles interact to improve our understanding of alcohol misuse. It also examines how different personality profiles are linked to distinct decision-making styles in cases of alcohol misuse. Given the significance of this topic, the study presents a large sample (including designs with and without NEO-PI-R), balanced by sex and age, and utilizes diverse assessment tools, demonstrating rigorous effort to clarify the impact of independent variables on alcohol misuse as a dependent variable.
However, I strongly recommend major revisions before the article can be considered for publication.
-
Subsection 1.1 begins with a confusing discussion of Hakulinen et al. (2015), featuring an overly long sentence that should be broken up for clarity. Later, while discussing alcohol misuse, the article states: “All this evidence confirms that substance abuse disorder has been strongly linked to personality traits such as high neuroticism, low conscientiousness, low agreeableness, and disinhibition (Kotov et al., 2010).” Since alcohol is a distinct substance with unique misuse patterns, these concepts should not be conflated.
-
Line 65: Move the comma in “disinhibition, impulsivity and, negative emotionality” so it reads: “disinhibition, impulsivity, and negative emotionality.”
-
Lines 67-69: “High levels of impulsivity and disinhibition would explain the capacity and willingness to adopt hasty, risky, and inappropriate behaviors or acts, such as alcohol misuse, which would result in unfavorable consequences (...).” Consequences such as what? Please specify.
-
Line 72: “actions, inhibit deadapted habitual responses”—This phrase is unclear. What is meant by “deadapted habitual responses”?
-
Line 78: Remove “in this case” from “Later, applying in this case the General Decision-Making Style.” It is unnecessary.
-
Lines 84-85: “They reported that considering decision-making styles as well as personality traits marginally improved the prediction of alcohol problems.” This sentence is confusing. Add missing commas for clarity.
-
Lines 89-92: The text states that procrastination and avoidant decision-making styles are linked to neuroticism and concludes that “deficits in control of negative emotions seem to play the main role in the association between decision-making styles and alcohol misuse.” This is an extrapolation. The claim needs stronger justification.
-
Lines 98-99: “For an excellent review, see Mezquita et al. (2021).” It is unusual to qualify a cited work as “excellent”, while also not summarizing its findings. Consider rewording.
-
Lines 99-101: “For instance, Zucker’s proposed a developmental model with four alcoholism subtypes: antisocial, developmentally limited, negative-affect, and primary (Zucker, 1994).” Instead, use: “Zucker (1994) proposed a developmental model with four alcoholism subtypes: antisocial, developmentally limited, negative-affect, and primary.” The citation at the end is redundant. Apply this correction throughout the article.
-
Lines 105-106: “This type has been associated with high harm avoidance (i.e., high neuroticism; García et al., 2012), and low novelty seeking.” Is low novelty seeking also from García et al. (2012)? If so, place the citation at the end; if not, add a separate reference.
-
Lines 124-128: The paragraph discussing the roles of disinhibition, positive emotionality, and negative emotionality in alcohol misuse is unclear and difficult to follow. Reword for clarity.
-
Lines 138-140: “Studies have suggested that personality traits and decision-making styles are related. Some traits present correlations between 0.40 and 0.60 with some decision-making styles (...).” Which traits? Avoid vague terms like “some” without specifying.
-
Method Section: The verb tenses are inconsistent. For example, it starts with “Two measures of alcohol misuse will be collected” but later states, “In this study, two impulsivity questionnaires were used.” Use past perfect consistently.
-
Scale descriptions: “The ZKA-PQ/SF is a short version of the ZKA-PQ that includes 80 items (four per facet) measuring five personality domains: Aggressiveness (AG), Activity (AC), Extraversion (EX), Neuroticism (NE), and Sensation Seeking (SS). This is an abbreviated version of the ZKA-PQ, which contains 200 items (Aluja et al., 2010).” The second sentence is redundant. Remove “This is an abbreviated version” since the first sentence already establishes that.
-
Barratt Impulsiveness Scale (BIS-11): The description does not specify what the response scale points (1-4) represent. Clarify.
-
General Decision-Making Style (GDMS) questionnaire: The explanation is convoluted. A clearer version:
“The GDMS is a self-administered instrument originally designed by Scott and Bruce (1995) with 25 items. Alacreu-Crespo et al. (2019) adapted it to Spanish, removing three items (resulting in 22 total) based on previous studies (e.g., Bavolar & Orosová, 2014) and their own analysis" -
Sampling method: “As a regular exercise, students had to administer a protocol containing the study instruments to seven people, including themselves.” Were student responses included in the sample? If so, does this raise ethical concerns? Clarify.
-
Lines 247-248: “Firstly, correlations of personality scales and decision-making styles with both alcohol misuse instruments will be computed.” Change “will be” to “were” for consistency.
-
Tables: The results tables are unclear. The text refers to “comparing low- and high-risk groups of alcohol misuse (AUDIT),” but the tables do not indicate where these groups are displayed (vertically? horizontally?). Improve readability.
-
Lines 355-357: “It should be highlighted that all pathways were related to alcohol misuse although they were uncorrelated.” What does this mean? Are the pathways uncorrelated with each other? Reword for clarity.
-
Lines 367-372: There are inconsistencies in font and letter formatting. Ensure uniformity throughout the text.
-
Lines 376-378: “This result was unexpected because Negative Emotionality or related traits have always been strongly associated with the continuous pattern of alcohol intake and abuse (Cloninger et al., 1996; Mezquita et al., 2021).” Remove “always” to maintain a more objective tone.
The article would strongly benefit from a revision by a native or highly proficient English speaker. Many sections are written in a confusing manner, suggesting that the authors may be thinking in their native language while writing.
For example:
- “Type I would be associated with positive reinforcement for its anti-anxiety effects, and rapidly developed tolerance and psychological dependence.”
The structure is awkward and unclear.
Additional examples and detailed comments are provided throughout the revision.
Author Response
Dear reviewer,
All authors appreciate your time dedicated to this review. Below you can find the specific answers to every comment.
- Subsection 1.1begins with a confusing discussion of Hakulinen et al. (2015), featuring an overly long sentence that should be broken up for clarity. Later, while discussing alcohol misuse, the article states: “All this evidence confirms that substance abuse disorder has been strongly linked to personality traits such as high neuroticism, low conscientiousness, low agreeableness, and disinhibition (Kotov et al., 2010).” Since alcohol is a distinct substance with unique misuse patterns, these concepts should not be conflated.
This way of introducing a brief summary of the association between alcohol and the FFM was inadequate, so we have completely rewritten the paragraph as follows: “One of the factors most associated with alcohol misuse has been personality traits (i.e., Aluja et al., 2019; Mezquita et al., 2021; Malouff et al., 2007; Soto, 2019). In a meta-analysis using the Five-Factor Model (FFM) as the personality framework, alcohol use disorder was associated with high neuroticism, low conscientiousness and low disinhibition (Kotov, et al., 2010). Later, and using an impressively large sample (72,949 adults), Hakulinen, et al., (2015) reported that high alcohol consumption was associated with high extraversion and low conscientiousness, whereas abstinence was associated with low extraversion, low openness and high agreeableness.” (lines 54-58).
- Line 65:Move the comma in “disinhibition, impulsivity and, negative emotionality” so it reads: “disinhibition, impulsivity, and negative emotionality.”
Corrected. Thanks.
- Lines 67-69:“High levels of impulsivity and disinhibition would explain the capacity and willingness to adopt hasty, risky, and inappropriate behaviors or acts, such as alcohol misuse, which would result in unfavorable consequences (...).” Consequences such as what? Please specify.
We have added the following information to this paragraph according to the references as follows: “which would result in unfavourable real-life consequences such as psychiatric disorders or economic deprivation (Ioannidis et al., 2019; Kovács et al., 2017).” (lines 64-65).
- Line 72:“actions, inhibit deadapted habitual responses”—This phrase is unclear. What is meant by “deadapted habitual responses”?
You are right. It could lead to confusion. The expression “deadapted habitual responses” has been deleted.
- Line 78:Remove “in this case” from “Later, applying in this case the General Decision-Making Style.” It is unnecessary.
Removed. Thanks
- Lines 84-85:“They reported that considering decision-making styles as well as personality traits marginally improved the prediction of alcohol problems.” This sentence is confusing. Add missing commas for clarity.
This sentence has been rewritten as follows: “They reported that decision-making styles marginally improved the prediction of alcohol problems reached by personality traits, but they also suggested that dependent and avoidant decision-making styles could play some role.” (lines 75-77).
- Lines 89-92:The text states that procrastination and avoidant decision-making styles are linked to neuroticism and concludes that “deficits in control of negative emotions seem to play the main role in the association between decision-making styles and alcohol misuse.” This is an extrapolation. The claim needs stronger justification.
We have added this phrase mainly to state that it is necessary to focus on this trait to understand the relationships among those decision-making styles and personality traits: “Convergent validity and theoretical studies show that Procastination (Heidari, & Arani, 2017; Urieta et al, 2021) and Dependent and Avoidant (Scott & Bruce, 1995, Urieta, et al., 2022) decision-making styles are mainly associated with the Neuroticism personality trait. Therefore, deficits in control of negative emotions (Mader et al., 2023) seem to play the main role in the association between decision-making styles and alcohol misuse”. (lines 79-81). We have added a reference (Mader et al., 2023) to remark that Neuroticism trait is related with deficits in the control of negative emotions.
Mader, N., Arslan, R. C., Schmukle, S. C., & Rohrer, J. M. (2023). Emotional (in) stability: Neuroticism is associated with increased variability in negative emotion after all. Proceedings of the National Academy of Sciences, 120(23), e2212154120.
- Lines 98-99:“For an excellent review, see Mezquita et al. (2021).” It is unusual to qualify a cited work as “excellent”, while also not summarizing its findings. Consider rewording.
We are aware that it is unusual, and we understand your view, but the present manuscript has been largely inspired by the work by Mezquita et al., (2021). Besides, arguments developed by Mezquita et al., (2021) are summarized in some parts of the paper such as, for instance: “Recently, Mezquita et al., (2021) proposed an alcohol developmental model. Interestingly, they explored the role of three broad personality domains (Disinhibition, Positive emotionality and Negative emotionality) in three different phases of alcohol intake (onset, use, misuse). Disinhibition would be largely involved in the three phases and plays a prominent role to account for use and misuse of alcohol. Positive emotionality would be associated with the beginning of alcohol consumption and specific patterns of heavy consumption, but its impact would be less influential in the development of long-term alcohol misuse, and, finally, negative emotionality would be mainly involved in alcohol misuse. This model is supported by some results that suggest that excessive alcohol consumption during adolescence may be partly driven by excitement seeking, but problematic use may be a consequence of disinhibition or an attempt to reduce negative mood (Stautz, & Cooper, 2013). (lines 107-114), or “Considering Mezquita et al´s (2021) model, it is expected that the disinhibition pathway will present the largest association with alcohol misuse, although reward sensitivity (i.e., sensation seeking) and negative emotionality (i.e. Neuroticism) will be associated with alcohol misuse as well.“ (lines 136-138). So, if you don’t mind, we have decided to retain this word.
- Lines 99-101:“For instance, Zucker’s proposed a developmental model with four alcoholism subtypes: antisocial, developmentally limited, negative-affect, and primary (Zucker, 1994).” Instead, use: “Zucker (1994) proposed a developmental model with four alcoholism subtypes: antisocial, developmentally limited, negative-affect, and primary.” The citation at the end is redundant. Apply this correction throughout the article.
We have corrected this here and elsewhere. Again, we appreciate your suggestion.
- Lines 105-106:“This type has been associated with high harm avoidance (i.e., high neuroticism; García et al., 2012), and low novelty seeking.” Is low novelty seeking also from García et al. (2012)? If so, place the citation at the end; if not, add a separate reference.
The reference García et al., (2012) at this point is no longer necessary. It has been deleted.
- Lines 124-128:The paragraph discussing the roles of disinhibition, positive emotionality, and negative emotionality in alcohol misuse is unclear and difficult to follow. Reword for clarity.
We totally agree. The paragraph in its present form is somewhat confusing. We have rewritten it as follows: Recently, Mezquita et al., (2021) proposed an alcohol developmental model. Interestingly, they explored the role of three broad personality domains (Disinhibition, Positive emotionality and Negative emotionality) in three different phases of alcohol intake (onset, use, misuse). Disinhibition would be largely involved in the three phases and plays a prominent role to account for use and misuse of alcohol. Positive emotionality would be associated with the beginning of alcohol consumption and specific patterns of heavy consumption, but its impact would be less influential in the development of long-term alcohol misuse, and, finally, negative emotionality would be mainly involved in alcohol misuse. This model is supported by some results that suggest that excessive alcohol consumption during adolescence may be partly driven by excitement seeking, but problematic use may be a consequence of disinhibition or an attempt to reduce negative mood (Stautz, & Cooper, 2013). (lines 107-114).
- Lines 138-140:“Studies have suggested that personality traits and decision-making styles are related. Some traits present correlations between 0.40 and 0.60 with some decision-making styles (...).” Which traits? Avoid vague terms like “some” without specifying.
We have specified three personality traits that mostly overlap with decision making styles. We have also deleted the word “some”. So, this sentence now reads as follows: “Some traits (such as Neuroticism, Conscientiousness, or Sensation Seeking) present correlations between 0.40 and 0.60 with decision-making styles (i.e., Heidari, & Arani, 2017; Rahaman, 2014; Urieta, et al., 2022)” (lines 21-122).
Note that we have not specified the relationships between every decision-making style and personality traits since this information is well-known and available in the references cited, and is not an aim of the present study.
- Method Section:The verb tenses are inconsistent. For example, it starts with “Two measures of alcohol misuse will be collected” but later states, “In this study, two impulsivity questionnaires were used.” Use past perfect consistently.
Thank you for this suggestion. Now, the past simple tense has been used throughout the method section.
- Scale descriptions:“The ZKA-PQ/SF is a short version of the ZKA-PQ that includes 80 items (four per facet) measuring five personality domains: Aggressiveness (AG), Activity (AC), Extraversion (EX), Neuroticism (NE), and Sensation Seeking (SS). This is an abbreviated version of the ZKA-PQ, which contains 200 items (Aluja et al., 2010).” The second sentence is redundant. Remove “This is an abbreviated version” since the first sentence already establishes that.
We agree. The paragraph has been simplified as follows: “Zuckerman–Kuhlman-Aluja Personality Questionnaire shortened form (ZKA-PQ/SF; Aluja et al., 2018). The ZKA-PQ/SF is a short version of the ZKA-PQ that measures five personality domains: Aggressiveness (AG), Activity (AC), Extraversion (EX), Neuroticism (NE), and Sensation seeking (SS). This is an abbreviated version (80 items) of the longer original ZKA-PQ (Aluja et al., 2010).” (lines 169-172).
- Barratt Impulsiveness Scale (BIS-11):The description does not specify what the response scale points (1-4) represent. Clarify.
The specific answer options have been added: “It was adapted to the Spanish cultural context by Oquendo et al., (2001). The answer format has a 4-point scale ranging from 1 to 4 (rarely/never, occasionally, often, almost always).” (Lines 178-179).
- General Decision-Making Style (GDMS) questionnaire:The explanation is convoluted. A clearer version:
“The GDMS is a self-administered instrument originally designed by Scott and Bruce (1995) with 25 items. Alacreu-Crespo et al. (2019) adapted it to Spanish, removing three items (resulting in 22 total) based on previous studies (e.g., Bavolar & Orosová, 2014) and their own analysis"
We really appreciate your time to clarify this sentence. The authors agree that your version is much better, and we have included it. Thank you very much.
- Sampling method:“As a regular exercise, students had to administer a protocol containing the study instruments to seven people, including themselves.” Were student responses included in the sample? If so, does this raise ethical concerns?
Note that it is already stated that the student her/himself also completed the protocol in the following sentence: “As a regular exercise, they had to administer a protocol containing the instruments analysed in the present study as well as other psychological instruments to seven people with the following characteristics: the student her/himself, one male and female aged between 18 and 30 years, one male and female aged between 31 and 50 years, and one male and female older than 51 years.” (Lines 206-208)
Furthermore, all participants were informed of the aims of the research and their rights. Research was approved by the corresponding ethical committee as stated in the following sentence: The research was part of a project authorized by the university of [blinded for review] ethics committee (Code: CEIC 2160). (line 209)
- Lines 247-248:“Firstly, correlations of personality scales and decision-making styles with both alcohol misuse instruments will be computed.” Change “will be” to “were” for consistency.
Modified.
- Tables:The results tables are unclear. The text refers to “comparing low- and high-risk groups of alcohol misuse (AUDIT),” but the tables do not indicate where these groups are displayed (vertically? horizontally?). Improve readability.
Sample sizes and cut-off scores to define low and high-risk groups are commented on in the method section in the following sentence: “The value of 0 indicates an abstainer who has never had any problems with alcohol. A score of 1 to 7 suggests low-risk consumption according to World Health Organization (WHO) guidelines. Scores from 8 to 15 suggest hazardous or harmful alcohol consumption and a score of 16 or more indicates the likelihood of alcohol dependence (moderate-severe alcohol use disorder). In the present sample, two groups were formed: low-risk (891 in the total sample and 406 in the subsample completing the NEO-PI-R) and high risk (hazardous and alcohol dependence groups; 97 in the total sample and 40 in the subsample completing the NEO-PI-R). (lines 156-161). Cohen’s d is included in table 2.
- Lines 355-357:“It should be highlighted that all pathways were related to alcohol misuse although they were uncorrelated.” What does this mean? Are the pathways uncorrelated with each other? Reword for clarity.
To clarify, we have added the expression “each other” after the word uncorrelated.
- Lines 367-372:There are inconsistencies in font and letter formatting. Ensure uniformity throughout the text.
We have reviewed the manuscript to try to homogenize the font and letter format.
- Lines 376-378:“This result was unexpected because Negative Emotionality or related traits have always been strongly associated with the continuous pattern of alcohol intake and abuse (Cloninger et al., 1996; Mezquita et al., 2021).” Remove “always” to maintain a more objective tone.
Removed.
Comments on the Quality of English Language
The article would strongly benefit from a revision by a native or highly proficient English speaker. Many sections are written in a confusing manner, suggesting that the authors may be thinking in their native language while writing.
For example:
- “Type I would be associated with positive reinforcement for its anti-anxiety effects, and rapidly developed tolerance and psychological dependence.”
The structure is awkward and unclear.
Additional examples and detailed comments are provided throughout the revision.
We agree that the manuscript needs a review of the English style. This sentence has been rewritten as well as other paragraphs in the paper. A native English speaker with many years of experience correcting scientific documents, and in particular psychology papers, has also reviewed the manuscript.
Reviewer 2 Report
Comments and Suggestions for Authors
1.The abstract should be divided into three parts: background, methods, and results, with a brief description of the statistical analysis methods used.
2.What is the innovative aspect of this study compared to other studies?
3.Is the sample characteristic (e.g., gender and age distribution) sufficiently representative? Is it clearly defined?
4.What are the inclusion and exclusion criteria in the research methods?
5.How valid and reliable is the measurement instrument used in this study?
6.What clinical implications do these conclusions have? Please provide specific application suggestions or propose potential future research directions.
- Use academic terminology and avoid colloquial or repetitive language.
- If a sentence is long and complicated, break it into shorter, simpler sentences to improve readability. Express ideas directly using clear language instead of complex grammar. For example, "Personality traits are key factors in understanding alcohol intake and misuse" and "Decision-making styles could also be involved" can be combined into more concise statements.
Author Response
First of all, thank you for your insightful comments. Below, you can find the answers and the amendments made to the manuscript.
1.The abstract should be divided into three parts: background, methods, and results, with a brief description of the statistical analysis methods used.
We have retained the abstract structure required by the journal in this special issue. However, we have rewritten some parts of the abstract to clarify it, and we have explicitly stated the two main statistical techniques used (Exploratory factor and regression analyses).
2.What is the innovative aspect of this study compared to other studies?
We think that the present paper is innovative in two main ways. The first is that it considers personality traits and decision-making styles simultaneously to understand alcohol misuse better. Since both kinds of constructs play a role in this topic and present a large overlap, it is necessary to test the specific contribution of each construct to the development of alcohol misuse. The second is the main aim of the present paper. Since people develop alcohol misuse for different reasons, it is essential to understand the different profiles (pathways) better to improve assessment and psychological treatments. Both ideas are included in the aims section and the discussion in the following paragraphs:
3.Is the sample characteristic (e.g., gender and age distribution) sufficiently representative? Is it clearly defined?
Note that the sample gathering procedure was designed to get as close a representation of the general sample as possible. Note that gender percentages are balanced (46.7% males and 53.3% females) and that mean age is 44.52. So, the mean age is much higher than would have been reported if the sample had been composed of university students only.
4.What are the inclusion and exclusion criteria in the research methods?
No inclusion or exclusion criteria were defined. However, students were told that protocol could not be applied to foreigners whose mother tongue was not Spanish, or to people that could not complete the instruments properly.
5.How valid and reliable is the measurement instrument used in this study?
Since reliability is a compulsory condition to properly run a research study using questionnaires, our first analysis was to compute descriptive statistics and reliability indexes (Cronbach’s alpha) for all scales. Results reported in table 1 show that scales present an adequate distribution and were reliable in the present sample.
6.What clinical implications do these conclusions have? Please provide specific application suggestions or propose potential future research directions.
We have made some recommendations based on the results of the paper. We have also expanded on these recommendations and suggested futures studies in the following paragraphs: According to the main aim of the present study, it is reinforced the idea of considering different pathways in the assess-ment and treatment of alcohol misuse (Cloninger et al., 1996) or substance abuse disorder (Verhuel and van den Brick, 2000). Psychological interventions to reduce alcohol misuse (Onrust, et al., 2016) would benefit from considering personality traits as trigger factors of alcohol use and, especially, abuse. Different personalities shed light on understanding the specific motives of individuals for consuming alcohol and developing behavioural problems related with this use (Hell, et al., 2022) and under-standing better the decision-making dynamics that lead to negative clinical conditions such as alcohol misuse or other substance abuse disorders (Lee, 2013).
From a preventive standpoint, personality could make a big contribution to detecting high-risk individuals and designing selective and more effective programs for every individual to reduce and prevent negative alcohol-related outcomes (Edalati, et al., 2019). In this way, detecting disinhibited personality profiles would be one of the keys to detecting and preventing long-term negative consequences of alcohol misuse. For instance, identifying situational triggers of a disinhibited behavior or cognitive distortions associated with, for instance, the reinforcement of aggressive behavior would avoid serious alcohol misuse and, in turn, reduce the impact of alcohol on behavior (Conrod, et al., 2006; Newton, et al., 2012). Focusing on these high-risk individ-uals would be a possibly more successful strategy to reduce the general alcohol use in a society and negative outcomes related to alcohol which, in turn, would reduce costs for society and healthcare (Davis-Stober, et al., 2019).
There are limitations of the present study which have implications for future studies. The cross-sectional study design is a limitation to disentangle the dynamic relationship between personality and decision-making styles and hence understand misuse. Longitudinal studies are needed to understand how a personality profile and decision-making styles combine over time to un-derstand the beginning of alcohol use and how, for some individuals, this use becomes problematic. Moreover, since the present manuscript focuses on alcohol misuse measures, we can draw no conclusion about the beginning of alcohol use, and it has been argued that different personality factors affect different phases of alcohol consumption (Mezquita et al., 2021). Another limitation is the way to define pathways. In the present study, only theoretical guided personality profiles have been defined. However, more accurate measures of the dynamics of alcohol use of individuals are necessary. For instance, a non-personality external assessment is required to demonstrate if heavy drink behaviour in one individual has the motivation of involvement in aggressive and disinhibited situations, stress reduction or simply fun seeking (Verhuel, & van den Brick, 2000). Finally, the present manuscript seeks to contribute to the design of successful interventions to reduce and prevent alcohol misuse. An essential line of research would be to test how psychological interventions customized according to personality pathways could improve the effectiveness of therapy focusing on reducing disinhibition or aggressive behaviours or negative mood states. (Lines 338-367)
Comments on the Quality of English Language
- Use academic terminology and avoid colloquial or repetitive language.
- If a sentence is long and complicated, break it into shorter, simpler sentences to improve readability. Express ideas directly using clear language instead of complex grammar. For example, "Personality traits are key factors in understanding alcohol intake and misuse" and "Decision-making styles could also be involved" can be combined into more concise statements.
We have reviewed the English style according to the reviewers’ suggestions. A native English speaker with long experience reviewing psychology research papers has also reviewed this second draft of the manuscript.
Reviewer 3 Report
Comments and Suggestions for Authors
The paper wishes to investigate the combined effects of personality traits and decision making styles on alcohol abuse
There are a number of major concerns related to the introduction and aims of the study.
- Authors list several traits having been related to alcohol misuse (AM), several theories about types of AM, several theories of pathways leading to AM, but without deducting their intentions and aims for the present study from this literature and without elucidating the novel aspect they expect from their study.
- They also do not put forward why and how pathways are expected to be related to traits and decision making nor do they explain which styles of decision making will be associated with which particular traits and pathways . In summary: The introduction is an additive listing instead of a deduction of hypotheses.
- By the way the pathway model by Verhuel and van den Brick , consists of a motivation for drinking (stress reduction) and two traits (reward sensitivity and impulsivity) and therefore only stress reduction as a motivation for drinking can be regarded as a separate aspect
- Clear hypotheses have to be put forward at the end of the introduction
- Methods: The motivation of individuals to participate in the study must be reported, because this is of great influence on the results.
- It does not become clear which benefit authors expect from using several scales for AM As well as for impulsivity and for several other personality traits. There has to be a rationale for this.
- There is also no rationale for performing an extra analysis for the subsample that had completed the NEO, since neuroticism and extraversion were represented in the other questionnaires
- Number of items, and reliabilities of subscales of decision making should be reported
- Statistical evaluations are not derived from study questions. For instance why do authors perform a factor analysis of pathways together with the other personality traits and why do they use the continuous variable of AM in addition to the high/low groups
- What principles are applied in selecting the steps in the hierarchical regression analyses
- Table 3: legend is required for the terms block 1 and block 2 and for the signs+ and –
- Lower parts in Tables 4 and 6 need terms for rows and columns ( what do the coefficients mean?)
- Legend of table 5 must explain which were the predictor variables, and what selection of scores is represented and what the signs + and – represent
- The discussion has to be completely rewritten after a more hypotheses guided statistical evaluation.
Author Response
All authors appreciate your time dedicated to this review. Below you can find the specific answers to each comment.
- Authors list several traits having been related to alcohol misuse (AM), several theories about types of AM, several theories of pathways leading to AM, but without deducting their intentions and aims for the present study from this literature and without elucidating the novel aspect they expect from their study.
The novel aspects of this study are twofold: To investigate different pathways to alcohol misuse that have been largely ignored in the literature, and to test the incremental validity of decision-making styles and personality traits to account for alcohol misuse differences. Both ideas are stated in several parts of the paper as follows: “The literature largely supports that personality traits and decision-making styles are associated with alcohol misuse, the former being one of the most important factors to understand predisposition, continuous use and serious negative real-life consequences of alcohol misuse. The particular nature of alcohol misuse also implies that an impulsive, wrong and biased decision-making style might be relevant to understanding alcohol misuse as well. In addition, previous studies have suggested that personality traits and decision-making styles are related. Some traits (such as Neuroticism, Conscientiousness, or Sensation Seeking) present correlations between 0.40 and 0.60 with decision-making styles (i.e., Heidari, & Arani, 2017; Rahaman, 2014; Urieta, et al., 2022) , and given the high overlapping between the two constructs, it makes sense to test how much incremental variance is added to the other kind of factors, and also to combine the two to try to predict alcohol misuse as accurately as possible. This is the first aim of the study.” (lines 117-124) or “Furthermore, it is compulsory to identify different pathways to alcohol misuse because not considering heterogeneity can severely bias its assessment and lead to ineffective treatment (García et al., 2024). Therefore, the second and main aim of the present paper is to describe different personality profiles (i.e. motivations) of alcohol misuse according to the different path-ways proposed by Verhuel and van den Brick (2000) and Mezquita et al., (2021), and to test how much predictive power deci-sion-making styles add for every profile. In order to describe different pathways as broadly and accurately as possible, several personality measures (Zuckerman’s personality model, impulsivity and FFM) were applied. Considering Mezquita et al´s (2021) model, it is expected that the disinhibition pathway will present the largest association with alcohol misuse, although reward sensitivity (i.e., sensation seeking) and negative emotionality (i.e. Neuroticism) will be associated with alcohol misuse as well.” (lines 131-138).
- They also do not put forward why and how pathways are expected to be related to traits and decision making nor do they explain which styles of decision making will be associated with which particular traits and pathways . In summary: The introduction is an additive listing instead of a deduction of hypotheses.
Note that there are references to which traits are related to which pathwaysin several paragraphs. We have rewritten them as follows to clarify this crucial point of the manuscript: In a review article, Verhuel and van den Brick (2000) identified three pathways to substance abuse disorder and addictive behavior: (I) Behavioural disinhibition; (II) stress reduction; and (III) reward sensitivity. Different traits underline these three pathways: Antisocial, impulsive, disinhibited and aggressive traits in the first case, neuroticism and anxiety in the second, and sensation seeking, reward dependence and extroversion in the third. The impulsivity and sensation seeking pathways, or the externalizing pathways, suggest that alcohol use forms part of a more general pattern of problematic or antisocial behavior, in some cases driven by fun-seeking. The second pathway, negative-affect regulation, also known as the self-medication or internalizing pathway, refers to drinking alcohol to decrease distress. It is important to highlight that these three pathways are expected to be independent, and the presence of one of them is enough to develop alcohol misuse. Hence, considering all three together is essential to understand the specific motivation of people with an alcohol misuse condition.
Recently, Mezquita et al., (2021) proposed an alcohol developmental model. Interestingly, they explored the role of three broad personality domains (Disinhibition, Positive emotionality and Negative emotionality) in three different phases of alcohol intake (onset, use, misuse). Disinhibition would be largely involved in the three phases and plays a prominent role to account for use and misuse of alcohol. Positive emotionality would be associated with the beginning of alcohol consumption and specific patterns of heavy consumption, but its impact would be less influential in the development of long-term alcohol misuse, and, finally, negative emotionality would be mainly involved in alcohol misuse. This model is supported by some results that suggest that excessive alcohol consumption during adolescence may be partly driven by excitement seeking, but problematic use may be a consequence of disinhibition or an attempt to reduce negative mood (Stautz, & Cooper, 2013). (Lines 97-114).
Besides, we have added some expected results to the aims section of the introduction. This section has now been rewritten as follows: This is the first aim of the study. It is expected that aggressiveness, impulsivity, sensation seeking, and neuroticism traits will be associated with alcohol misuse. Moreover, decision-making styles will add some incremental validity beyond personality traits but not much (Bavolar and Bačíkova-Slešková, 2018). What is difficult to predict is the specific decision-making styles that will play a relevant role given the inconsistencies in the previous literature, where different studies (Bavolar and Bačíkova-Slešková, 2018; Bavolar, & Orosova, 2014; Phillips and Ogeil, 2011) have emphasized the role of different decision-making styles.
Furthermore, it is compulsory to identify different pathways to alcohol misuse because not considering heterogeneity can severely bias its assessment and lead to ineffective treatment (García et al., 2024). Therefore, the second and main aim of the present paper is to describe different personality profiles (i.e. motivations) of alcohol misuse according to the different path-ways proposed by Verhuel and van den Brick (2000) and Mezquita et al., (2021), and to test how much predictive power deci-sion-making styles add for every profile. In order to describe different pathways as broadly and accurately as possible, several personality measures (Zuckerman’s personality model, impulsivity and FFM) were applied. Considering Mezquita et al´s (2021) model, it is expected that the disinhibition pathway will present the largest association with alcohol misuse, although reward sensitivity (i.e., sensation seeking) and negative emotionality (i.e. Neuroticism) will be associated with alcohol misuse as well. (lines 124-138).
- By the way the pathway model by Verhuel and van den Brick, consists of a motivation for drinking (stress reduction) and two traits (reward sensitivity and impulsivity) and therefore only stress reduction as a motivation for drinking can be regarded as a separate aspect
One relevant point of the manuscript is to analyze the different pathways to alcohol misuse, since it essential to consider all of them to better understand the particular motivations of each person with alcohol misuse problems. On the other hand, it is necessary to state that the three pathways are independent since if they presented a large overlap , it would not make sense to analyze the three. This idea is now in the introduction section in these paragraphs: “It is important to highlight that these three pathways are expected to be independent, and the presence of one of them is enough to develop alcohol misuse. Hence, considering all three together is essential to understand the specific motivation of people with an alcohol misuse condition.” (lines 103-105).
Finally, note that the results of the present study suggest the independence of the three pathways, since the three factors were orthogonal.
- Clear hypotheses have to be put forward at the end of the introduction
As has been commented on in point 2, we have added some expected results to the aims section of the introduction. This section has been rewritten as follows: “It is expected that aggressiveness, impulsivity, sensation seeking, and neuroticism traits will be associated with alcohol misuse. Moreover, decision-making styles will add some incremental validity beyond personality traits but not much (Bavolar and Bačíkova-Slešková, 2018). What is difficult to predict is the specific decision-making styles that will play a relevant role given the inconsistencies in the previous literature, where different studies (Bavolar and Bačíkova-Slešková, 2018; Bavolar, & Orosova, 2014; Phillips and Ogeil, 2011) have emphasized the role of different decision-making styles” and “Considering Mezquita et al´s (2021) model, it is expected that the disinhibition pathway will present the largest association with alcohol misuse, although reward sensitivity (i.e., sensation seeking) and negative emotionality (i.e. Neuroticism) will be associated with alcohol misuse as well. (lines 124-138).
- Methods: The motivation of individuals to participate in the study must be reported, because this is of great influence on the results.
We have included a sentence about this point as follows: “No reward was given for completing the protocol, but, to increase motivation, scores on some personality traits were returned to all participants.” (lines 210-211). Note that participants were not rewarded, but a personality profile was given to increase motivation.
- It does not become clear which benefit authors expect from using several scales for AM As well as for impulsivity and for several other personality traits. There has to be a rationale for this.
Including many and varied personality measures is essential to properly describe the three pathways to alcohol misuse. This idea is expressed in the following sentence: “In order to describe different pathways as broadly and accurately as possible, several personality measures (Zuckerman’s personality model, impulsivity and FFM) were applied.” (lines 135-136).
Regarding the inclusion of two measures of alcohol misuse, we must recognize that there is not a clear rationale behind the decision. This research is within the framework of a wider research project about risky behaviours. As two measures of alcohol misuse were collected in this project, we used both of them. In any case, it is interesting that no sharp differences are reported between them, which amounts to a replication of results. We have included this idea on in two sentences in the results section.
- There is also no rationale for performing an extra analysis for the subsample that had completed the NEO, since neuroticism and extraversion were represented in the other questionnaires
The reviewer is right to state that Neuroticism and Extraversion are also assessed by ZKA-PQ/SF, but using one neuroticism scale precludes a proper definition of a Negative affectivity pathway in the total sample (table 4). Hence, to properly define the three pathways raised by Verhuel and van den Brick (2000) and Mezquita et al., (2021), another measure of Neuroticism must be included. In this case, the NEO-PI-R was also analysed in a subsample. Note that the inclusion of another scale of Neuroticism allows us to define a Negative Emotionally factor (Table 6). This idea is expressed in the following sentence: “It is possible that the Neuroticism factor was not identified in the total sample because there is one scale associated with this trait only, so the same analysis was conducted on the subsample including the NEO-PI-R to better identify the Neuroticism factor (i.e., the negative emotionally pathway).” (lines 230-232).
This solution also allows us to distinguish between aggressiveness and neuroticism, which is another contribution of the paper, and is analysed in the present paragraph of the discussion section: “Finally, a factor of Neuroticism presents a lower predictive validity of alcohol misuse. This result was unexpected because Negative Emotionality or related traits have been strongly associated with the continuous pattern of alcohol use and misuse (Cloninger et al., 1996; Mezquita et al., 2021). Results of the present study could indicate that the association between Negative Emotionality and alcohol misuse might be due to antisocial aspects of Neuroticism such as anger (Jones et al., 2011) more than Anxiety itself. This argument would agree with Aluja et al.`s (2019) results, where Anxiety was not predictive of alcohol drink-ing, and with the fact that the association with alcohol misuse was higher for the Neuroticism scales of the NEO-PI-R than for the corresponding scale of the ZKA-PQ/SF. Since the Neuroticism domain of the NEO-PI-R includes facets of depression or anger (Costa, & McCrae, 1999), this pattern of results would indicate that alcohol misuse as a self-medication strategy would have other psychological motivations beyond stress reduction. It could be considered a strategy to reduce any negative emotion, such as sadness, anger or undesired aggressivity (Fein, & Nip, 2012).” (lines 306-315).
- Number of items, and reliabilities of subscales of decision making should be reported
Reliabilities of the decision-making scales are reported in table 1. The number of items has now been included in the method section: “It was structured by five different domains, each representing a decision-making style (number of items between brackets): Rational (5), Intuitive (3), Dependent (5), Avoidant (5), and Spontaneous (4).” (lines 193-195).
- Statistical evaluations are not derived from study questions. For instance why do authors perform a factor analysis of pathways together with the other personality traits and why do they use the continuous variable of AM in addition to the high/low groups
A factor analysis is compulsory to operationalize the personality pathways and to compute factor scores (i.e., individual level) separately for the different pathways. As stated in the introduction, pathways are personality profiles, so they have to be described after personality traits. In regard to AM, depending on the nature of the analysis (linear or non-linear), AM measures have been used as continuous variable (regression analyses), or used to define two groups in accordance with the literature about AUDIT and to compute Cohen’s d.
- What principles are applied in selecting the steps in the hierarchical regression analyses
We have conducted two hierarchical analyses. In the first one, decision-making styles are entered in the first block, and personality traits in the second. Later, the blocks are reversed (table 3). This analysis answers the question of what incremental validity each type of variable adds, especially the decision-making styles when they are entered in the second block.
The second hierarchical regression analysis is similar but considering the pathways in the first block, and the decision-making styles in the second (tables 5 and 7). In this case, the aim is to test how much variance is added by the decision-making styles for every pathway.
- Table 3: legend is required for the terms block 1 and block 2 and for the signs+ and –
Blocks are the steps of the hierarchical regression analysis. This is explained in the text. A sentence about the significance of the +/- signs has been added to the legends.
- Lower parts in Tables 4 and 6 need terms for rows and columns ( what do the coefficients mean?)
The meaning of the lower parts of table 4 and 6 is stated in the title of the table as follows: “correlations among factors, and Cohen’s d (last row) for the factor scores comparing low and high-risk groups of alcohol misuse (AUDIT).”
- Legend of table 5 must explain which were the predictor variables, and what selection of scores is represented and what the signs + and – represent
The predictor variables are the three pathways/profiles (block 1), and the five decision-making styles (Block 2). We have added a sentence to clarify this point.
- The discussion has to be completely rewritten after a more hypotheses guided statistical evaluation.
The discussion section has been largely rewritten in line with the changes to the introduction section and to the answers to some of the reviewer’s comments. The main changes can be seen in these paragraphs: “The present study largely supports the definition of different profiles/pathways (i.e., motivations) to alcohol misuse. It should be highlighted that all pathways were related to alcohol misuse although they were uncorrelated. It therefore largely reinforces the need to explore different pathways to alcohol misuse. As we also expected, the antisocial/disinhibition pathway is the most important profile to understand misuse, which clearly supports the developmental model put forward by Mezquita et al., (2021). People that tend to be aggressive and disrespectful of social norms would have the highest risk of developing alcohol misuse and the corresponding negative outcomes in real life (Bjork, et al., 2004; Mezquita et al., 2021). Sensation seeking, meanwhile, would be the second most relevant pathway to alcohol misuse. In this case, motivation to consume could be attributable to appetitive motivation. This is also in agreement with the fact that enhancement drinking motives (i.e., “to get a high” or “because it’s fun”) have been associated with different alcohol-related outcomes, such as drinking quantity and frequency. The relevance of Sensation Seeking suggests that personality models other than the Big-Five (which only considered Sensation Seeking as an Extroversion facet in the gold standard of the FFM – NEO-PI-R; Costa, & McCrae, 1999) should be considered, in particular Zuckerman’s model, in which sensation seeking is a basic dimension of human personality (Zuckerman et al., 2005). Note that this trait has been related to biases in decision-making (Reynolds et al., 2019), especially with unplanned and risky behaviours with negative consequences for the person and the group (Ioannidis et al., 2019; Kovács et al., 2017). Impulsivity, which is factorially and psychologically different from sensation seeking (García et al., 2012), also plays a role, albeit lesser than suggested in other studies (Acton, 2003).
Finally, a factor of Neuroticism presents a lower predictive validity of alcohol misuse. This result was unexpected because Negative Emotionality or related traits have been strongly associated with the continuous pattern of alcohol use and misuse (Cloninger et al., 1996; Mezquita et al., 2021). Results of the present study could indicate that the association between Negative Emotionality and alcohol misuse might be due to antisocial aspects of Neuroticism such as anger (Jones et al., 2011) more than Anxiety itself. This argument would agree with Aluja et al.`s (2019) results, where Anxiety was not predictive of alcohol drinking, and with the fact that the association with alcohol misuse was higher for the Neuroticism scales of the NEO-PI-R than for the corresponding scale of the ZKA-PQ/SF. Since the Neuroticism domain of the NEO-PI-R includes facets of depression or anger (Costa, & McCrae, 1999), this pattern of results would indicate that alcohol misuse as a self-medication strategy would have other psychological motivations beyond stress reduction. It could be considered a strategy to reduce any negative emotion, such as sadness, anger or undesired aggressivity (Fein, & Nip, 2012).
The results support the hypothesis that, beyond personality traits, decision-making styles make a small contribution to understanding alcohol misuse (usually lower than 5%; Bavolar and Bačíkova-Slešková, 2018). In the present study, Spontaneous style plays the main role (as it does in Bavoloar & Orosova, 2014). It should be highlighted that this decision-making style increases the predictive power irrespective of the profiling (Goudriaan, et al., 2007). Thus, the desire to resolve a situation quickly, though in an inappropriate or maladaptive way, puts a person at greater risk of developing alcohol misuse beyond the specific personality pathway. The main role of spontaneous style is also in agreement with the fact that it is involved in other risky behaviours such as driving unsafely (Aluja et al., 2023). In regard to the other decision-making styles, avoidant style also seems to be relevant (Bavolar, & Bačíkova-Slešková, 2018), but it is surprising that rational or intuitive, which have been associated with other negative mental health outcomes (Aluja et al., 2023; Bavolar, & Bačíkova-Slešková, 2020), played no role, although this pattern has already been replicated in previous studies (Bavolar, & Bačíkova-Slešková, 2018; Bavoloar & Orosova, 2014).
There are limitations of the present study which have implications for future studies. The cross-sectional study design is a limitation to disentangle the dynamic relationship between personality and decision-making styles and hence understand misuse. Longitudinal studies are needed to understand how a personality profile and decision-making styles combine over time to understand the beginning of alcohol use and how, for some individuals, this use becomes problematic. Moreover, since the present manuscript focuses on alcohol misuse measures, we can draw no conclusion about the beginning of alcohol use, and it has been argued that different personality factors affect different phases of alcohol consumption (Mezquita et al., 2021). Another limitation is the way to define pathways. In the present study, only theoretical guided personality profiles have been defined. However, more accurate measures of the dynamics of alcohol use of individuals are necessary. For instance, a non-personality external assessment is required to demonstrate if heavy drink behaviour in one individual has the motivation of involvement in aggressive and disinhibited situations, stress reduction or simply fun seeking (Verhuel, & van den Brick, 2000). Finally, the present manuscript seeks to contribute to the design of successful interventions to reduce and prevent alcohol misuse. An essential line of research would be to test how psychological interventions customized according to personality pathways could improve the effectiveness of therapy focusing on reducing disinhibition or aggressive behaviours or negative mood states.
Round 2
Reviewer 1 Report
Comments and Suggestions for Authors
The authors have made a clear effort to address all the review comments. However, I would prefer that the adjective "excellent" be removed. A phrase like "for a comprehensive review" would be more suitable, in my opinion, but I leave that to the authors' discretion. Additionally, I strongly believe that the responses from those who had applied the questionnaire should have been excluded from the final analysis. However, since this was not an issue for the Ethics Committee, I also leave the final decision to the authors.
Author Response
All authors appreciate your positive impression about the first review. Your suggestion about using “for a comprehensive review” is fine with us, and we have used it. In regard to excluding the responses from those who had applied the questionnaires, we have decided to retain them because psychometric information supports the proper application of all questionnaires in the entire sample (alpha coefficients were similar to original studies and no relevant biases were detected). Besides, they are about a quarter of the sample, so removing them would make the sample more restrictive. I hope you understand our decision.
Reviewer 3 Report
Comments and Suggestions for Authors
Although authors have taken great effort to improve the manuscript, it still suffers from heavy lack of conceptual clarity and consequent reasoning:
1.—The major trouble is: When developing Aims of the manuscript there is a mixture of the terms “pathways into AM, and personality traits .Pathways are partly identical with personality traits ( as in the Verhuel model (disinhibition and reward dependence) while the third one (stress reduction) is a drinking motive , which of course is related to the trait of Neuroticism/ Harm Avoidance
Because of this inaccurate differentiation the first aim (lines 124- 139) and what is declared as aim 2 ( line 132 ) are practically identical because aim 2 repeats the questions of aim 1 with respect to pathways, (which however, are combinations of the personality traits listed above
Furthermore, aim 1 is a mixture of two questions ( predicting AM from traits and incremental validity of decision making styles) .
Since the relations of the traits to AM have been thoroughly described in previous literature , the only way to make Aim1 useful, would be to ask: which trait contributes most to AM
As a consequence of listing different traits published as essential correlates of alcohol misuse, these should have been selected from respective personality inventories and included into a factor analysis as performed later and then the newly developed factor scores should be related to AM.
A logical issue for developing the aims and statistical analyses would be:
Question 1. Which traits contribute most to predicting AM scores (correlations and multiple
regression)
Question 2. Do factors of relevant traits mirror the pathways into AM as depicted by the Verhuel and
VandenBrick model? ( first step: factor analysis, then correlation of factor scores with AM scores)
Question 3. a)Relationship of decision making styles to AM and b) incremental validity of dec.mak
styles in addition to personality factors (simple correlations and hierarchical regression with
delta R² for dec.mak st.)
More critical details:
- --Line 55 FFM does not contain disinhibition as a factor
3--Line 71ff : for readers not familiar with sub-styles of decision making a short explanation should be given for dependent and avoidant and in line 82 vigilant style
4.--How do the Cloninger or the Verhuel and van den Brick Models relate to the questionnaires used for AM
5.--What is the use of presenting the developmental model , if authors do not use it in their AM questionnaires and do not intend to relate phases to the traits and decision styles Otherwise Justification has to be given for choosing the Verhuel model among other models for Question 2 (see my new list of questions)
6---Lines 117-123 are redundant,( already outlined in the paragraph above)
7.--Lines 90 ff: After personality traits have been reasonably related to pathways the reader expects that different pathways are going to be investigated, but instead, the instruments used for AM only differentiate between light and heavy alcohol drinking, and subgroups are formed according to this dimension.
8.--The novel question about incremental validity of decision making should include the question which types of decision making are best suitable for adding variance to predicting AM from the known personality traits
9.--Methods: What type of “linear analysis” was performed?
10.--The question how high and low risks´ drinkers differ, and which personality traits best predict AM has not been raised at the end of the introduction
11.-- Line 226 according to Verhuel and Van den Brick the three expected pathway factors should be impulsivity( Disinhibition, Neuroticism and Reward Sensitivity) , not the ones listed in line 227
12 --Table 3 What do the two halves of the table represent ? This should appear in the title of the table,´
13.--furthermore, what is meant by incremental validity of the personality traits?incrementak to which analysis?
14.--Table 5 does not explain what “ blocks” refer to
Author Response
Once again, we really appreciate your time in reviewing this second version of the manuscript.
1.—The major trouble is: When developing Aims of the manuscript there is a mixture of the terms “pathways into AM, and personality traits .Pathways are partly identical with personality traits ( as in the Verhuel model (disinhibition and reward dependence) while the third one (stress reduction) is a drinking motive , which of course is related to the trait of Neuroticism/ Harm Avoidance
Because of this inaccurate differentiation the first aim (lines 124- 139) and what is declared as aim 2 ( line 132 ) are practically identical because aim 2 repeats the questions of aim 1 with respect to pathways, (which however, are combinations of the personality traits listed above
Furthermore, aim 1 is a mixture of two questions ( predicting AM from traits and incremental validity of decision making styles) .
Since the relations of the traits to AM have been thoroughly described in previous literature , the only way to make Aim1 useful, would be to ask: which trait contributes most to AM
As a consequence of listing different traits published as essential correlates of alcohol misuse, these should have been selected from respective personality inventories and included into a factor analysis as performed later and then the newly developed factor scores should be related to AM.
A logical issue for developing the aims and statistical analyses would be:
Question 1. Which traits contribute most to predicting AM scores (correlations and multiple
regression)
Question 2. Do factors of relevant traits mirror the pathways into AM as depicted by the Verhuel and
VandenBrick model? ( first step: factor analysis, then correlation of factor scores with AM scores)
Question 3. a)Relationship of decision making styles to AM and b) incremental validity of dec.mak
styles in addition to personality factors (simple correlations and hierarchical regression with
delta R² for dec.mak st.)
- We agree that the way the three aims are introduced is still unconvincing (point 1 of your review). So, according to your suggested order and amendments, we have rewritten the aims section. Regarding one of your observations, however, the third and main aim of the present paper has been retained as the third one because we think that the aims relating to relationships between personality traits and decision-making styles (aim 1) and incremental validity (aim 2) should be together. Now, this section reads as follows: “The literature largely supports that personality traits and decision-making styles are associated with alcohol misuse, the former being one of the most important factors to understand predisposition, continuous use and serious negative real-life consequences of alcohol misuse. So, the first aim is to replicate the reported relationships between alcohol misuse and personality traits and decision-making styles. It is expected that aggressiveness, impulsivity, sensation seeking, and neuroticism traits will be associated with alcohol misuse. What is difficult to predict is the specific decision-making styles that will play a relevant role given the inconsistencies in the previous literature, where different studies (Bavolar and Bačíkova-Slešková, 2018; Bavolar, & Orosova, 2014; Phillips and Ogeil, 2011) have emphasized the role of different decision-making styles. On the other hand, previous studies have suggested that personality traits and decision-making styles are related. Some traits (such as Neuroticism, Conscientiousness, or Sensation Seeking) present correlations between 0.40 and 0.60 with decision-making styles (i.e., Heidari, & Arani, 2017; Rahaman, 2014; Urieta, et al., 2022), and given the high overlapping between the two constructs, it makes sense to test how much incremental variance is added to the other kind of factors, and also to combine the two to try to predict alcohol misuse as accurately as possible. This is the second aim of the study. It is hypothesized that decision-making styles will add some incremental validity beyond personality traits, but not much (Bavolar and Bačíkova-Slešková, 2018).
Furthermore, it is compulsory to identify different pathways to alcohol misuse because not considering heterogeneity can severely bias its assessment and lead to ineffective treatment (García et al., 2024). Therefore, the third and main aim of the present paper is to describe different personality profiles (i.e. motivations) of alcohol misuse according to the different pathways proposed by Verhuel and van den Brick (2000) and Mezquita et al., (2021), and to test how much predictive power decision-making styles add for every profile. In order to describe different pathways as broadly and accurately as possible, several personality measures (Zuckerman’s personality model, impulsivity and FFM) were applied. Considering Mezquita et al´s (2021) model, it is expected that the disinhibition pathway will present the largest association with alcohol misuse, although reward sensitivity (i.e., sensation seeking) and negative emotionality (i.e. Neuroticism) will be associated with alcohol misuse as well.” (lines 117-138).
2--Line 55 FFM does not contain disinhibition as a factor
2.- You are right because FFM does not contain disinhibition as a factor, but we have retained it because the original reference uses this concept explicitly.
3--Line 71ff : for readers not familiar with sub-styles of decision making a short explanation should be given for dependent and avoidant and in line 82 vigilant style
- Note that a definition of the five decision-making styles assessed in the present manuscript is already included in the method section (lines 191-200). On the other hand, we have not included a definition of vigilant style because this concept is barely used in the present manuscript.
4.--How do the Cloninger or the Verhuel and van den Brick Models relate to the questionnaires used for AM
- The associations of Cloninger’s traits and the three pathways suggested by Verhuel and van den Brick are commented on in the following paragraphs: “Several different profiles of alcohol use and misuse have been described (for a comprehensive review, see Mezquita et al., 2021). For instance, Zucker (1994) proposed a developmental model with four alcoholism subtypes: antisocial, developmentally limited, negative-affect, and primary. Cloninger devised a well-known typology of alcoholism that has inspired much research on the topic (Cloninger, et al., 1996). He suggested the presence of two types of alcoholics (I and II). Type I is characterized mainly by the motivation to reduce tension and anxiety, which results in a rapidly developed tolerance and psychological dependence. This type has been associated with the trait poles of high harm avoidance (i.e., high neuroticism), and low novelty seeking. On the other hand, Type II was associated with positive reinforcement for its euphoriant and stimulant effects because of their natural need of stimulation. This second type would present an antisocial and sensation seeking profile. These and other typologies and classifications clearly suggest that people develop a substance disorder, addiction or alcohol misuse due to different motivations and for different, mostly unrelated, etiologies (Dawe, & Loxton, 2004).
In a review article, Verhuel and van den Brick (2000) identified three pathways to substance abuse disorder and addictive behavior: (I) Behavioural disinhibition; (II) stress reduction; and (III) reward sensitivity. Different traits underline these three pathways: Antisocial, impulsive, disinhibited and aggressive traits in the first case, neuroticism and anxiety in the second, and sensation seeking, reward dependence and extroversion in the third. The impulsivity and sensation seeking pathways, or the externalizing pathways, suggest that alcohol use forms part of a more general pattern of problematic or antisocial behavior, in some cases driven by fun-seeking. The second pathway, negative-affect regulation, also known as the self-medication or internalizing pathway, refers to drinking alcohol to decrease distress. It is important to highlight that these three pathways are expected to be independent, and the presence of one of them is enough to develop alcohol misuse. Hence, considering all three together is essential to understand the specific motivation of people with an alcohol misuse condition.” (lines 86-105). Note that not all Cloninger`s traits have been linked to alcohol misuse. The first paragraph focused on the two main ones: Harm Avoidance and Novelty Seeking. In regard to Verheul and van den Brick (2000), the personality roots of the three pathways are described.
5.--What is the use of presenting the developmental model , if authors do not use it in their AM questionnaires and do not intend to relate phases to the traits and decision styles Otherwise Justification has to be given for choosing the Verhuel model among other models for Question 2 (see my new list of questions)
- Mezquita et al.,’s (2021) developmental model associated the three phases of alcohol intake (onset, use, misuse) with three broad personality domains (Disinhibition, Positive emotionality and Negative emotionality). Note that we focus on the third phase since we measured alcohol misuse only. In this sense, expected results focus on the alcohol misuse phase of Mezquita et al.,’s (2021) model, as stated in the following sentence: “Considering Mezquita et al´s (2021) model, it is expected that the disinhibition pathway will present the largest association with alcohol misuse, although reward sensitivity (i.e., sensation seeking) and negative emotionality (i.e. Neuroticism) will be associated with alcohol misuse as well” (line. 136-138). Finally, we have used Verhuel and van den Brick (2000) and Mezquita et al.,`s (2021) models because they accurately described different personality pathways to alcohol misuse and, in our opinion, and are essential to guide treatment and prevention, as stated in the following sentence: “Furthermore, it is compulsory to identify different pathways to alcohol misuse because not considering heterogeneity can severely bias its assessment and lead to ineffective treatment (García et al., 2024).” (lines 131-132).
6---Lines 117-123 are redundant,( already outlined in the paragraph above)
- As has been stated above, your criticisms of the aims section are fair, and we have rewritten them.
7.--Lines 90 ff: After personality traits have been reasonably related to pathways the reader expects that different pathways are going to be investigated, but instead, the instruments used for AM only differentiate between light and heavy alcohol drinking, and subgroups are formed according to this dimension.
- Note that the three different pathways have been statistically operationalized after exploratory factor analysis. In fact, the extracted factors resemble quite well the three pathways. Later, individual levels on those factors (that is to say, risk of developing alcohol misuse due to this personality profile) was associated with alcohol misuse. On the other hand, differences between low and high-risk groups were also associated with pathways to offer a complementary non-linear analysis.
8.--The novel question about incremental validity of decision making should include the question which types of decision making are best suitable for adding variance to predicting AM from the known personality traits
- We agree that the hypotheses should include which decision-making styles would add more variance to personality traits. However, previous evidence lacks replicability about this specific topic. Different studies support the role of different decision-making styles. Since no clear hypothesis about which decision-making styles would play the main role can be made, we have added the following sentence in regard to aim 1: “What is difficult to predict is the specific decision-making styles that will play a relevant role given the inconsistencies in the previous literature, where different studies (Bavolar and Bačíkova-Slešková, 2018; Bavolar, & Orosova, 2014; Phillips and Ogeil, 2011) have emphasized the role of different decision-making styles.” (lines 121-123). Note that this sentence could be applied to the second aim as well.
9.--Methods: What type of “linear analysis” was performed?
- Both correlations and regression analyses were conducted to test the linear relationships between personality traits and decision-making styles with alcohol misuse.
10.--The question how high and low risks´ drinkers differ, and which personality traits best predict AM has not been raised at the end of the introduction
- Now this question is more clearly covered in the aims section. Note that we have not raised specific hypothesis for both different analyses (Low vs High-risk drinkers and linear analyses).
11.-- Line 226 according to Verhuel and Van den Brick the three expected pathway factors should be impulsivity( Disinhibition, Neuroticism and Reward Sensitivity) , not the ones listed in line 227
- You are right, but this is because of the nature of the variables considered in the factor analysis. In any case, the deviation from the first pathway (Disinhibition) is slight since this pathway also includes antisocial and aggressive traits, as stated in lines 97-99: “In a review article, Verhuel and van den Brick (2000) identified three pathways to substance abuse disorder and addictive behavior: (I) Behavioural disinhibition; (II) stress reduction; and (III) reward sensitivity. Different traits underline these three pathways: Antisocial, impulsive, disinhibited and aggressive traits in the first case”.
12 --Table 3 What do the two halves of the table represent ? This should appear in the title of the table,´
- We think that the current title reflects quite well the nature of the analysis reported in table 3. Note that two hierarchical regression analyses were conducted. In the first one, decision-making styles were introduced in the first block and personality traits in the second. In the second, personality traits were entered in the first block, and decision-making styles in the second. Note that the subheadings are quite accurate about this point.
13.--furthermore, what is meant by incremental validity of the personality traits?incrementak to which analysis?
- Incremental analysis means what amount of predictive variance the variables entered in the second block add. It refers to hierarchical regression analyses (tables 3, 5 and 7).
14.--Table 5 does not explain what “ blocks” refer to
- Block refers to the steps of the hierarchical regression analyses. In the case of Table 5, it is described in the results section: “When the regression analyses including the factor scores as independent variables are conducted (table 5), the percentage of variance is about 10% for Aggressiveness/Neuroticism, 7% for Sensation Seeking and 3% for Impulsivity. It seems that the antisocial pathway presents the largest correlation with alcohol misuse. It should be remarked that decision-making styles increment the predictive validity by about 5%, the most important styles being once again Avoidant and Spontaneous. Like the previous analysis, no sharp differences are reported between results for AUDIT or RAPI.” (Lines 266-270).
Round 3
Reviewer 3 Report
Comments and Suggestions for Authors
Thanks for replying to each of the amendments,
However, the manuscript still lacks a strict logical structure.-Therefore I would request the following further steps for revision of the present version of the manuscript:
- Please separate the three questions in the introduction by separate lines
- Please arrange the results in the sequence according to the questions by separate headlines
- Most importantly, emphasize in the introduction when defining pathways, that pathways actually address personality traits ( e.g. behavioural disinhibition see subscale of Zuckerman´s sensation seeking battery ,and reward dependence see the questionnaire by Torrubia, and theories by Gray; Eysenck;)
- Please structure presentation of results in the order of questions 1,2,3 with respective section headlines.
5 .There is no rationale for presenting a separate analysis for the subsample with data on the NEO-PI-R. Extraversion and Neuroticism are covered by the other scales available from the total sample. Therefore, please eliminate the extra factor analysis and hierarchical regression analysis based on the NEO-PI –R and passages 277-285 and adjust the abstract and respective results and discussion passages accordingly. The fact that by this analysis, aggression and neuroticism were split into two factors could at best be touched in the discussion, mentioning a separate analysis, because this is not a major question at the onset of the study.
Author Response
Reviewer 3
Again, we appreciate your time devoted to the paper. We agree that the logical structure of the aims and results section could be improved. We have made the following modifactions:
- Please separate the three questions in the introduction by separate lines
We have separated the paragraphs about the first and second aims. Thus, the paragraph about the second aim has been rewritten as follows: “The second aim focuses on incremental validity. Previous studies have suggested that personality traits and decision-making styles are related. Some traits (such as Neuroticism, Conscientiousness, or Sensation Seeking) present correlations between 0.40 and 0.60 with decision-making styles (i.e., Heidari, & Arani, 2017; Rahaman, 2014; Urieta, et al., 2022). Hence, given the high overlapping between the two constructs, we will test how much incremental variance is added to the other kind of construct, and also combine the two to try to predict alcohol misuse as accurately as possible. It is hypothesized that decision-making styles will add some incremental validity beyond personality traits, but not much (Bavolar and Bačíkova-Slešková, 2018). (lines 125-130).
- Please arrange the results in the sequence according to the questions by separate headlines
We have added three headings for each analysis and the corresponding aim:
3.1. Association of alcohol misuse with personality traits and decision-making styles
3.2. Incremental validity of personality traits and decision-making styles to account for alcohol misuse.
3.3. Pathways to alcohol misuse: Profiling, prediction and incremental validity of decision-making styles
- Most importantly, emphasize in the introduction when defining pathways, that pathways actually address personality traits ( e.g. behavioural disinhibition see subscale of Zuckerman´s sensation seeking battery ,and reward dependence see the questionnaire by Torrubia, and theories by Gray; Eysenck;)
We agree that it is important to remark that pathways are defined using personality traits, so we have added the expression “pathways address personality traits” when they are introduced in the text: “It is important to highlight that these three pathways address personality traits, and are expected to be independent, the presence of one of them being enough to develop alcohol misuse. Hence, considering all three together is essential to understand the specific motivation of people with an alcohol misuse condition.” (103-105).
- Please structure presentation of results in the order of questions 1,2,3 with respective section headlines.
As stated in point 2, three headings have been added to the results section.
5 .There is no rationale for presenting a separate analysis for the subsample with data on the NEO-PI-R. Extraversion and Neuroticism are covered by the other scales available from the total sample. Therefore, please eliminate the extra factor analysis and hierarchical regression analysis based on the NEO-PI –R and passages 277-285 and adjust the abstract and respective results and discussion passages accordingly. The fact that by this analysis, aggression and neuroticism were split into two factors could at best be touched in the discussion, mentioning a separate analysis, because this is not a major question at the onset of the study.
The present manuscript analyses two frameworks about alcohol misuse (Mezquita et al., 2021; Verhuel, & van den Brick, 2000). Both of them include a negative affectivity or stress reduction pathway. In fact, this inclusion is in strong agreement with the fact that negative emotion scales, such as neuroticism, are associated with alcohol use and misuse. On the other hand, note that pathways should be defined as factors, not scales. They are psychological constructs (i.e., latent factors) associated with, in this case, alcohol misuse. The first analysis with the total sample did not properly identify the negative affectivity or stress reduction pathways because one neuroticism scale is included only. No other scale about negative emotions, anxiety or similar is analyzed. So, to better or more precisely identify stress reduction or negative affectivity pathways of both frameworks, it is necessary to add other neuroticism or negative emotions scales. This is why we have analyzed the NEO-PI-R on a subsample. This idea is commented on in the following sentence: “It is possible that the negative affectivity factor was not identified in the total sample because there is one scale associated with this trait only (Neuroticism scale of the ZKA-PQ/SF). Hence, the same analysis was conducted on the subsample including the NEO-PI-R to better identify the negative emotionality or stress reduction pathway”. (lines 231-232).
When the NEO-PI-R is added, the negative affectivity pathway is identified correctly (Neuroticism factor; table 6). So, considering the relevant role played by this pathway or profile regarding alcohol misuse (as stated in the introduction section), we think that this analysis is essential to the aims of the paper, and we have decided to retain it. We hope you understand our reasons. Besides, it should be remarked that this analysis has no effect on the rest of the paper and is merely complementary. In any case, we have added a sentence to reinforce the necessity of including more negative affectivity scales as follows: “It is also essential to remark that the inclusion of one neuroticism scale only precludes the identification of stress reduction (Verhuel, & van den Brick, 2000) or Negative emotionality (Mezquita et al., 2021) pathways. In fact, in this solution the Neuroticism scale of the ZKA-PQ/SF loaded on the first factor (Antisocial/aggressiveness).” (lines 266-269).